# Behavioral correlates of cortical semantic representations modeled by word vectors

**Satoshi Nishida**[1,2]*, **Antoine Blanc**[1], **Naoya Maeda**[3], **Masataka Kado**[3], **Shinji Nishimoto**[1,2,4]

**1** Center for Information and Neural Networks (CiNet), National Institute of Information and Communications Technology (NICT), Suita, Osaka, Japan, **2** Graduate School of Frontier Biosciences, Osaka University, Suita, Osaka, Japan, **3** NTT DATA Corporation, Tokyo, Japan, **4** Graduate School of Medicine, Osaka University, Suita, Osaka, Japan

* s-nishida@nict.go.jp

**Data Availability Statement:** The data underlying the results presented in the study are available from GitHub (https://github.com/s-nishida/behav_corr_sem_model). The raw behavioral data collected in the word-arrangement task are

## Abstract

The quantitative modeling of semantic representations in the brain plays a key role in understanding the neural basis of semantic processing. Previous studies have demonstrated that word vectors, which were originally developed for use in the field of natural language processing, provide a powerful tool for such quantitative modeling. However, whether semantic representations in the brain revealed by the word vector-based models actually capture our perception of semantic information remains unclear, as there has been no study explicitly examining the behavioral correlates of the modeled brain semantic representations. To address this issue, we compared the semantic structure of nouns and adjectives in the brain estimated from word vector-based brain models with that evaluated from human behavior. The brain models were constructed using voxelwise modeling to predict the functional magnetic resonance imaging (fMRI) response to natural movies from semantic contents in each movie scene through a word vector space. The semantic dissimilarity of brain word representations was then evaluated using the brain models. Meanwhile, data on human behavior reflecting the perception of semantic dissimilarity between words were collected in psychological experiments. We found a significant correlation between brain model- and behavior-derived semantic dissimilarities of words. This finding suggests that semantic representations in the brain modeled via word vectors appropriately capture our perception of word meanings.

## Author summary

Word vectors, which have been originally developed in the field of engineering (natural language processing), have been extensively leveraged in neuroscience studies to model semantic representations in the human brain. These studies have attempted to model brain semantic representations by associating them with the meanings of thousands of words via a word vector space. However, there has been no study explicitly examining whether the brain semantic representations modeled by word vectors actually capture our perception of semantic information. To address this issue, we compared the semantic

available from Open Science Framework (https://osf.io/um3qg/).

**Funding:** The work was supported by Japan Society for the Promotion of Science (https://www.jsps.go.jp/english/) KAKENHI Grant-in-Aid for Early-Career Scientists (18K18141) to S.N. and for Young Scientists A (15H05311) to S.N., and Japan Science Technology agency (https://www.jst.go.jp/EN/) PRESTO (JPMJPR20C6) to S.N. and ERATO (JPMJER1801) to S.N.. The funders had no role in study design, data collection and analysis, decision to publish, or preparation of the manuscript.

**Competing interests:** I have read the journal's policy and the authors of this manuscript have the following competing interests: This study was funded by NTT Data Corp. NM and MK are employees of NTT Data Corp.

representational structure of words in the brain estimated from word vector-based brain models with that evaluated from behavioral data in psychological experiments. The results revealed a significant correlation between these model- and behavior-derived semantic representational structures of words. This indicates that the brain semantic representations modeled using word vectors actually reflect the human perception of word meanings. Our findings contribute to the establishment of word vector-based brain modeling as a useful tool in studying human semantic processing.

## Introduction

Natural language processing is a branch of machine learning that aims to develop machines that understand the meanings of words. In the field of natural language processing, a number of algorithms have been developed to capture the semantic representations of words from word statistics in large-scale text data as word vectors [1–5]. The word vectors obtained using these algorithms have effectively captured the latent semantic structure of words and further performed various types of natural language tasks, such as word similarity judgment [3,4,6], sentiment analysis [7,8], and question answering [9].

Furthermore, word vectors can be also used in neuroimaging studies to model semantic representations in the brain [10–17]. These studies have reported that word vector-based models have the ability to predict the brain response evoked by semantic perceptual experiences [10,12–16]. These models are also able to recover perceived semantic contents from brain response [11,17,18]. These findings suggest that word vectors capture at least some aspects of the semantic representations in the brain. However, whether the brain semantic representations modeled by word vectors accurately reflect the semantic perception of humans is yet to be determined. In other words, no study has yet identified the behavioral correlates of the modeled brain semantic representations. This clarification is important in order to establish the brain modeling with word vectors as an accurate methodology for investigating human semantic processing.

To examine the behavioral correlates of the brain semantic representations modeled by word vectors, we compared the semantic representational structure estimated by word vector-based brain models with that evaluated from human behavior. The estimation of the semantic representational structure in the brain was performed using voxelwise modeling with a word vector space of fastText [4], GloVe [3], or word2vec [2] (Fig 1A). For this purpose, we conducted two sets of functional magnetic resonance imaging (fMRI) experiments in which 52 participants were asked to view natural movies in the scanner; among them, 36 participated in one or the other of these experiments, and 16 participated in both. The word vector-based voxelwise models predicted movie-evoked fMRI signals from the semantic contents in individual movie scenes in the word vector space. We then transformed original word vectors into brain representations using the model weights, and a brain-derived word dissimilarity matrix from these brain representations was obtained (Fig 1B).

Meanwhile, the behavior-derived semantic representational structure was obtained from behavioral data in a psychological task, in which participants separately arranged tens of words (nouns and adjectives) in a two-dimensional space as per their semantic relationship (Fig 2). This task was a modified version of a psychological task introduced previously [19]. A behavior-derived word dissimilarity matrix was then estimated using these behavioral data. Finally, we examined the correlation between the brain- and behavior-derived word dissimilarity matrices separately for both nouns and adjectives.

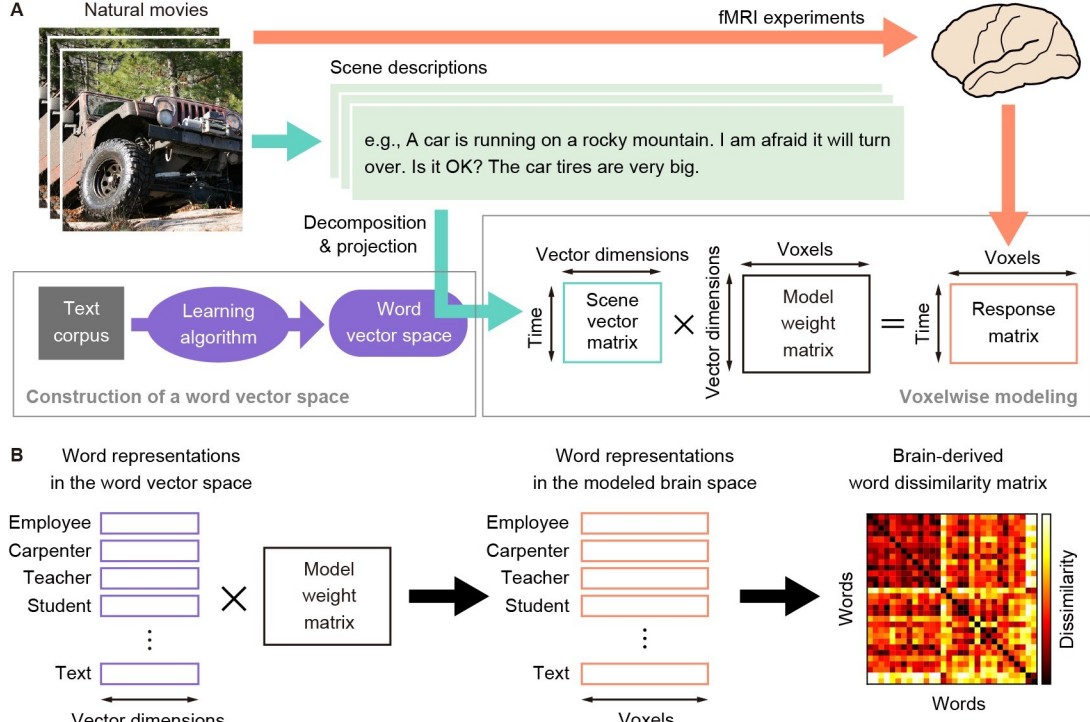

**Fig 1. Voxelwise modeling and brain-derived word dissimilarity. A**) Voxelwise modeling based on a word vector space. The model predicts fMRI signals evoked by natural movie scenes via a weighted linear summation of the vector representations of semantic descriptions of each scene (scene vectors). Scene vectors were obtained by transforming manual descriptions of each movie scene through a word vector space pretrained using statistical learning (fastText, GloVe, or word2vec) from a text corpus. The weights of the linear prediction model were trained using the corresponding time series of movies and fMRI signals of each brain. **B**) Estimation of brain-derived word dissimilarity. Word representations in the vector space were transformed into word representations in the modeled brain space by multiplying original word vectors by the model weights. Then, the correlation distance of the modeled word representations between all possible pairs of words was calculated, producing a brain-derived word dissimilarity matrix.

## Results

### Performance of voxelwise models based on word vectors

We first determined whether the voxelwise models based on a word vector space appropriately predict movie-evoked brain responses. The performance of brain-response prediction was

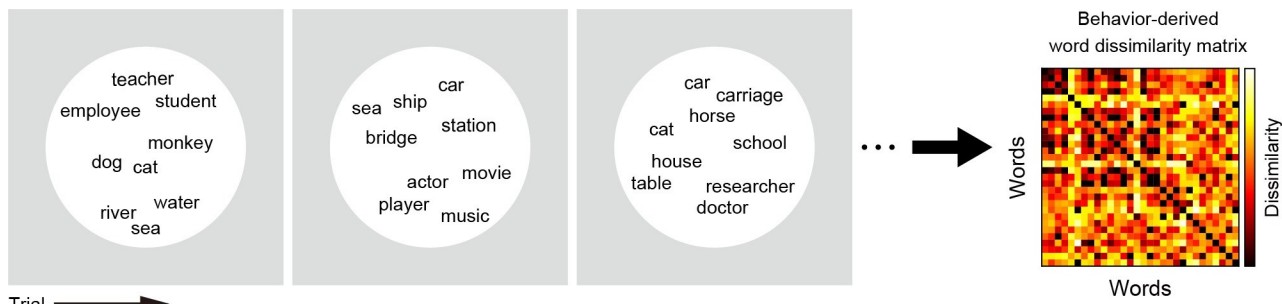

**Fig 2. Word-arrangement task and behavior-derived word dissimilarity.** To evaluate the word dissimilarity structure derived from human behavior, we conducted psychological experiments in which each participant performed a word-arrangement task. On each trial of this task, participants were required to arrange multiple ($\leq$60) words in a two-dimensional space according to the semantic relationship of those words. After each participant completed $\leq$1 h of this task, a behavior-derived word dissimilarity matrix was established using inverse multidimensional scaling (see Methods for more details).

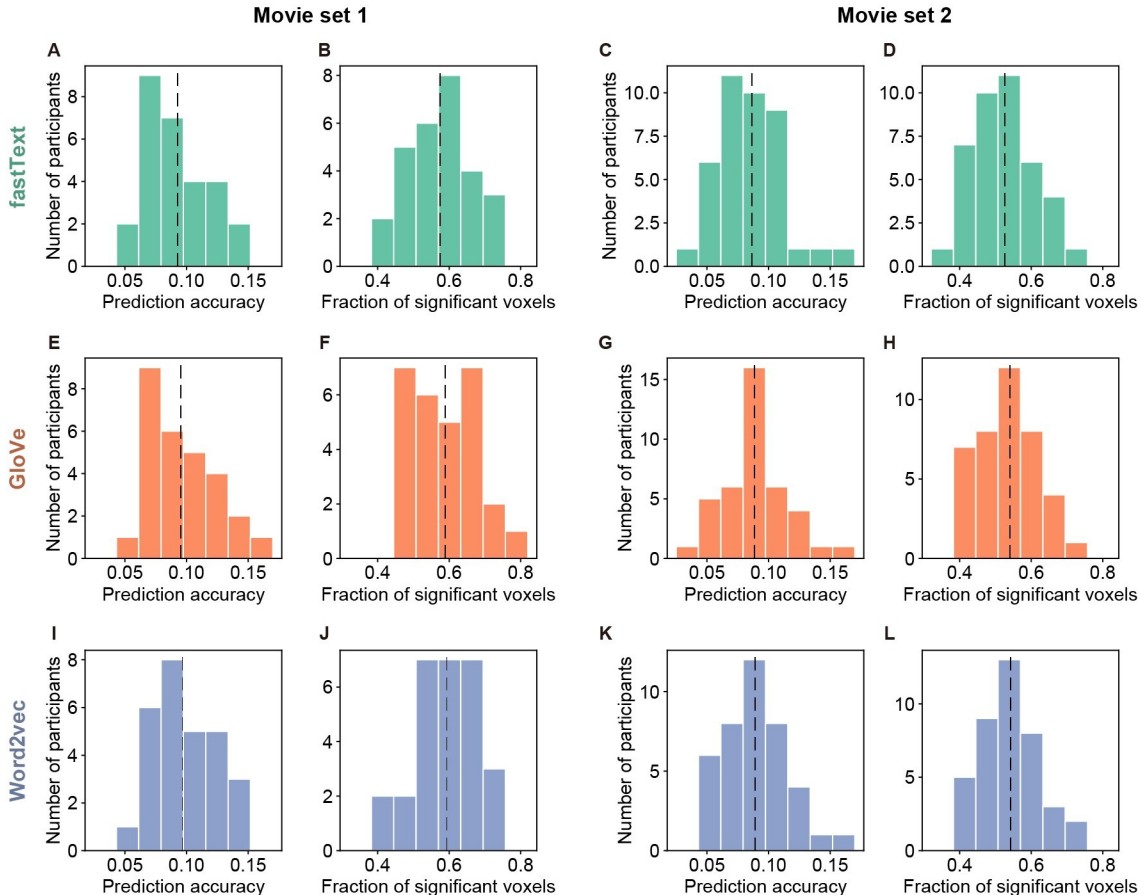

**Fig 3. Performance of voxelwise models in brain-response prediction.** The performance of voxelwise models (vector dimension = 1000) was evaluated in terms of predicting brain responses in the test dataset. For this purpose, prediction accuracy and the fraction of significant voxels for each brain were calculated for each type of word vectors (**A–D**, fastText; **E–H**, GloVe; **I–L**, word2vec). The distribution of prediction accuracy (**A, C, E, G, I**, and **K**) and that of the fraction of significant voxels (**B, D, F, H, J**, and **L**) were separately shown for movie sets 1 (**A–B, E–F**, and **I–J**) and 2 (**C–D, G–H**, and **K–L**). The vertical dashed line in each panel indicates the prediction accuracy averaged across participants.

evaluated using the following two measures: (1) prediction accuracy calculated as the correlation coefficients between predicted and measured brain responses in the test dataset, and (2) the fraction of significant voxels, among all cortical voxels, for which the prediction accuracy reached a significance threshold ($p < 0.05$ after the correction for multiple comparisons using the false discovery rate [FDR]; for more details, see Methods). The prediction accuracy averaged across all cortical voxels and across participants for the fastText, GloVe, and word2vec vector spaces was 0.0928, 0.0955, and 0.0967, respectively, for movie set 1 (28 participants) and 0.0865, 0.0884, and 0.0891, respectively, for movie set 2 (40 participants); no participant showed mean prediction accuracy less than 0 (Fig 3A, 3C, 3E, 3G, 3I and 3K). This accuracy was sufficiently high because it was averaged over all cortical voxels, and previous studies on voxelwise modeling have reported a similar tendency [20–22]. The fraction of significant voxels averaged across participants for the fastText, GloVe, and word2vec vector spaces was 0.575, 0.589, and 0.594, respectively, for movie set 1 and 0.526, 0.541, and 0.543, respectively, for movie set 2 (Fig 3B, 3D, 3F, 3H, 3J and 3I). These results indicate that the voxelwise models trained showed sufficient performance in the modeling of semantic representations consistently across different stimulus sets.

In the comparison of prediction performance between different types of word vectors, word2vec vectors showed significantly higher accuracy than the other vectors for both movie sets (Wilcoxon test, $p < 0.05$, FDR corrected) while GloVe vectors showed significantly higher accuracy than fastText vectors for both movie sets ($p < 0.0001$, FDR corrected). In the comparison of the fraction of significant voxels, word2vec and GloVe vectors showed significantly higher fractions than fastText vectors for both movie sets ($p < 0.0001$, FDR corrected) while the differences between word2vec and GloVe vectors were not significant for both movie sets ($p > 0.06$, FDR corrected). Nonetheless, the differences of the prediction performance between these vectors were totally small ($<0.004$ for mean prediction accuracy and $<0.02$ for the mean fraction of significant voxels).

Although the vector dimensionality of the word vector spaces had little effect on the model performance, there was no clear tendency toward vector-dimensionality dependency. The change of model performance was not monotonic against the change of vector dimensions (Kendall rank correlation, $\tau = -0.071$–$0.708$, $p > 0.17$) and was inconsistent across movie sets and different vector types (S1–S3 Figs).

Previous studies performed voxelwise modeling using simpler, discrete word features, such as binary labels of objects and actions occurring in movie scenes (e.g., [23]). However, because such discrete word features have been examined separately from word vectors in voxelwise modeling, it is still unclear whether word vectors are more effective to model semantic representations in the brain than discrete word features. To address this, we compared these two types of semantic features, first of all, in terms of the performance of brain-response prediction. For this purpose, we constructed a voxelwise model using the binary labeling of semantic contents in each movie scene (binary-labeling model). Although this model was similar to one used in a previous study [23], we obtained binary labels of semantic contents by extracting words from the manual scene descriptions rather than manual word labeling performed in the previous study (for more details, see Methods). In this model, a semantic feature in a given scene was represented by a binary vector with 100–2000 dimension that corresponded to the occurrence of 100–2000 specific words in manual descriptions in that scene.

We found that the binary-labeling model constructed for each participant showed sufficiently high performance in brain-response prediction especially when high-dimensional vectors were used (S4 Fig). For 1000 and 2000 vector dimensions, the performance of the binary-labeling model was rather significantly higher than that of all the word vector-based models (Wilcoxon test, $p < 0.05$, FDR corrected); nonetheless, the performance differences between models were not large ($<0.006$ for mean prediction accuracy and $<0.05$ for the mean fraction of significant voxels). This result suggests that brain response to semantic information in natural scenes can be effectively predicted even using discrete word features.

Does this finding imply that the semantic relational structure of words, captured by word vectors [3,4,6], is ineffective in the modeling of cortical semantic representations? Comparing the word vector-based and binary-labeling model, however, cannot purely assess the effectiveness of semantic relational structure of word vectors because the procedures to transform scene descriptions to semantic features were different between these models. To examine the effectiveness more fairly, we compared the performance of brain-response prediction between the voxelwise model based on original, trained word vectors and that based on untrained word vectors. The same procedure as used for original word vectors was employed to obtain the untrained word vectors, except that the statistical training of word vectors was omitted. This procedure yielded 1000-dimensional random vectors, each assigned to each word in the vocabulary shared with original word vectors. Hence, the untrained vectors still had signatures of individual words but not the semantic relational structure of words at all. Then, the

voxelwise model based on the untrained word vectors were constructed using the same procedure as used for the voxelwise model based on original, trained word vectors (see also Methods).

The performance of brain-response prediction (i.e., prediction accuracy and the fraction of significant voxels) was compared between the voxelwise model based on trained vectors and that based on untrained vectors. We found that the prediction performance of voxelwise models was consistently higher for trained vectors than for untrained vectors regardless of performance measures, word-vector types, and movie sets (S5 Fig). This result suggests that the semantic relational structure of words, captured by trained word vectors, is effective in the prediction of movie-evoked brain response, namely, in the modeling of cortical semantic representations.

## Localization of cortical regions highly predicted by voxelwise models

We next identified which cortical regions were predictable by the word vector-based models in order to determine that the models could capture the semantic representations in cortical regions which are considered to be involved in audiovisual semantic processing. For this purpose, the prediction accuracy averaged across participants was calculated in each of the 148 cortical regions that were anatomically segmented using FreeSurfer [24,25] on the basis of the Destrieux atlas [26]. Then, the averaged prediction accuracy for each region was mapped onto the cortical surface of a reference brain (Fig 4 and Table 1). The highly predictable regions for each word vector-based model were localized over widespread cortical regions, including the occipital, superior and inferior temporal, and posterior parietal regions. This localization was consistent with previous reports, in which the semantic representations in the brain were modeled with word features [12,23]. There was a strong correlation of mean prediction accuracy over 148 cortical regions between different types of word vectors (Pearson's r > 0.999; Spearman's $\rho$ > 0.999) and between movie sets 1 and 2 (Pearson's r > 0.972; Spearman's $\rho$ > 0.973); this indicates the consistency of model predictability of these cortical regions across vector types and stimulus sets. In addition, there were similar tendencies across vector dimensionality (S6–S8 Figs and S1–S6 Tables). These results indicate that the word vector-based voxelwise models trained reliably showed the localization of highly predictable regions.

These results showed that the mean prediction accuracy in each brain region ranged up to 0.37 across word-vector types and movie sets. It might be argued that the prediction accuracy observed in this study was deemed much lower than that observed in previous studies (up to 0.6–0.8) [21,23]. However, this discrepancy is likely to be explained by the difference in spatial granularity used for calculating prediction accuracy; namely, we showed brain region-wise prediction accuracy whereas the previous studies reported voxelwise prediction accuracy. At the voxel level, our voxelwise models, including binary-labeling models, for individual brains exhibited prediction accuracy ranging up to ~0.7 (S9 and S10 Figs), which is consistent with the results from the previous studies [21,23].

A similar localization was also observed when the fraction of significant voxels in each cortical region was mapped onto the cortical surface (Fig 5 and Table 2). The fraction was relatively large in the occipital, superior and inferior temporal, and posterior parietal regions compared with the other regions. This localization pattern was observed to be highly consistent across vector types, movie sets, and vector dimensions (S11–S13 Figs and S7–S12 Tables). Taken together, these results suggest that the word vector-based models capture the semantic representations in appropriate cortical regions.

It should be noted that, however, the fraction of significant voxels was not small even in the other cortical regions, including the prefrontal cortex. The minimum fraction within any

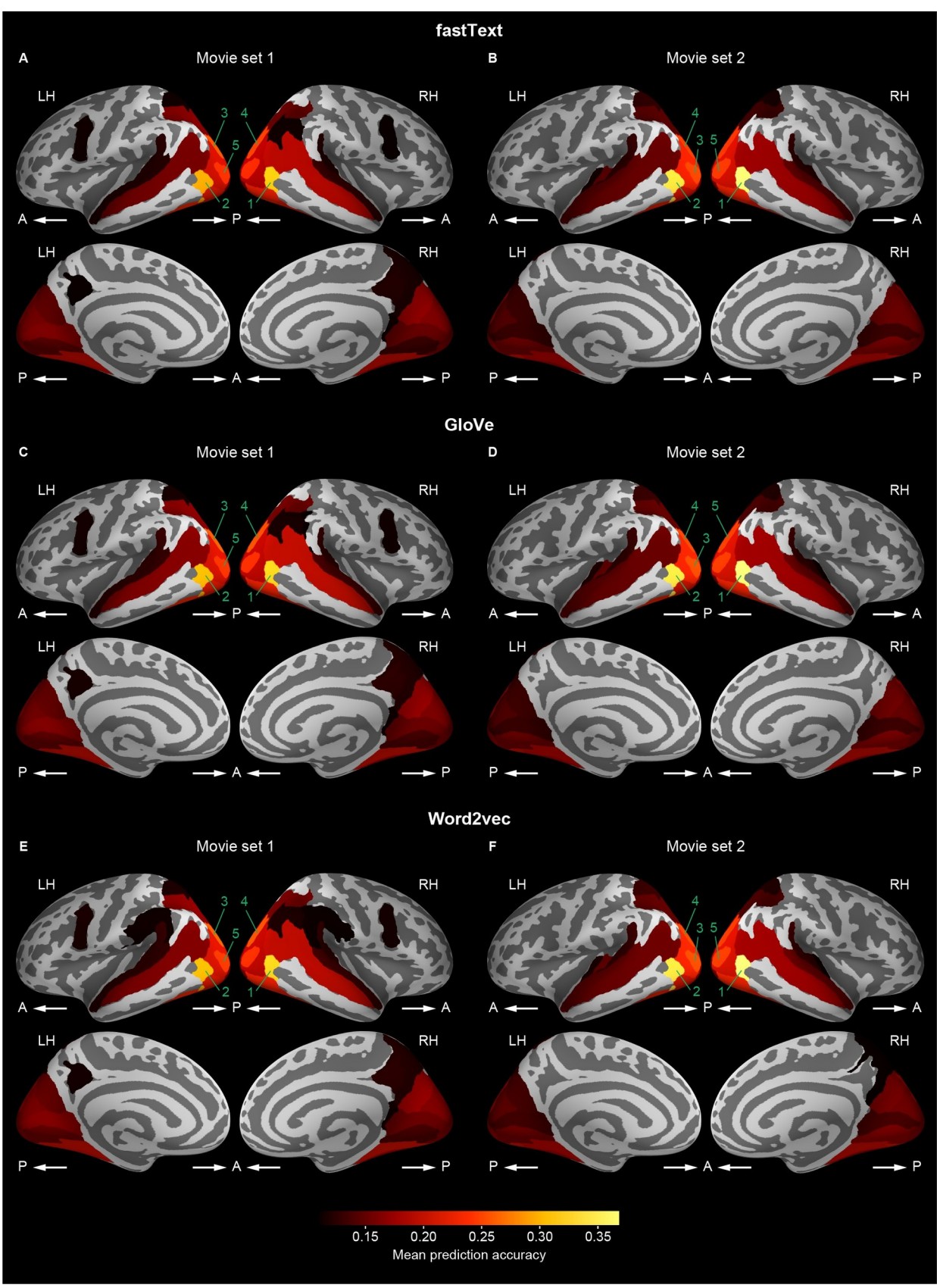

**Fig 4. Cortical mapping of prediction accuracy.** Participant-averaged prediction accuracy of voxelwise models (vector dimension = 1000) was mapped onto the cortical surface of a reference brain for each of fastText (**A** and **B**), GloVe (**C** and **D**), and word2vec (**E** and **F**) vectors and for each of movie sets 1 (**A**, **C**, and **E**) and 2 (**B**, **D**, and **F**). The prediction accuracy was averaged within each of the cortical regions that were anatomically segmented. Brighter colors in the surface maps indicate cortical regions that have higher prediction accuracy. We showed only regions with mean prediction accuracy above 0.11, which reaches a significance level (i.e., p = 0.05) of prediction accuracy after Bonferroni correction for multiple comparisons among 148 cortical regions (i.e., p = 0.0001 ~ 0.05/148). The five cortical regions with the highest mean prediction accuracy are numbered in a descending order separately for each type of word vectors and for each movie set. The names of these regions are shown in Table 1. LH, left hemisphere; RH, right hemisphere; A, anterior; P, posterior.

individual region was 0.237, which is significantly higher than 0.05 (corresponding to the significance level of 0.05; Wilcoxson test, p < 0.00001). In addition, even when the chance level of the fraction for each region was estimated from control voxelwise models in which word vectors were shuffled across vector dimensions for each vector (see also Methods), the fraction of significant voxels was significantly above the chance level in all the cortical regions (Wilcoxson test, p < 0.00001, FDR corrected; S14 Fig). These results suggest that although high prediction accuracy was observed in specific regions (Figs 4 and S6–S8), the word vector-based model can potentially capture semantic information even from other regions across the cortex.

**Table 1. Cortical regions with the highest mean prediction accuracy for each type of word vectors and for each movie set.**

|  |  | Rank | Region name | Prediction accuracy |
|---|---|---|---|---|
| fastText | Movie set 1 | 1 | Right anterior occipital sulcus and preoccipital notch | 0.312 |
|  |  | 2 | Left anterior occipital sulcus and preoccipital notch | 0.293 |
|  |  | 3 | Left superior occipital sulcus and transverse occipital sulcus | 0.254 |
|  |  | 4 | Right superior occipital sulcus and transverse occipital sulcus | 0.245 |
|  |  | 5 | Left middle occipital sulcus and lunatus sulcus | 0.238 |
|  | Movie set 2 | 1 | Right anterior occipital sulcus and preoccipital notch | 0.365 |
|  |  | 2 | Left anterior occipital sulcus and preoccipital notch | 0.337 |
|  |  | 3 | Left middle occipital sulcus and lunatus sulcus | 0.248 |
|  |  | 4 | Left superior occipital sulcus and transverse occipital sulcus | 0.245 |
|  |  | 5 | Right middle occipital sulcus and lunatus sulcus | 0.242 |
| GloVe | Movie set 1 | 1 | Right anterior occipital sulcus and preoccipital notch | 0.320 |
|  |  | 2 | Left anterior occipital sulcus and preoccipital notch | 0.301 |
|  |  | 3 | Left superior occipital sulcus and transverse occipital sulcus | 0.258 |
|  |  | 4 | Right superior occipital sulcus and transverse occipital sulcus | 0.251 |
|  |  | 5 | Left middle occipital sulcus and lunatus sulcus | 0.241 |
|  | Movie set 2 | 1 | Right anterior occipital sulcus and preoccipital notch | 0.368 |
|  |  | 2 | Left anterior occipital sulcus and preoccipital notch | 0.340 |
|  |  | 3 | Left middle occipital sulcus and lunatus sulcus | 0.251 |
|  |  | 4 | Left superior occipital sulcus and transverse occipital sulcus | 0.249 |
|  |  | 5 | Right superior occipital sulcus and transverse occipital sulcus | 0.244 |
| Word2vec | Movie set 1 | 1 | Right anterior occipital sulcus and preoccipital notch | 0.319 |
|  |  | 2 | Left anterior occipital sulcus and preoccipital notch | 0.302 |
|  |  | 3 | Left superior occipital sulcus and transverse occipital sulcus | 0.260 |
|  |  | 4 | Right superior occipital sulcus and transverse occipital sulcus | 0.252 |
|  |  | 5 | Left middle occipital sulcus and lunatus sulcus | 0.243 |
|  | Movie set 2 | 1 | Right anterior occipital sulcus and preoccipital notch | 0.367 |
|  |  | 2 | Left anterior occipital sulcus and preoccipital notch | 0.339 |
|  |  | 3 | Left middle occipital sulcus and lunatus sulcus | 0.251 |
|  |  | 4 | Left superior occipital sulcus and transverse occipital sulcus | 0.249 |
|  |  | 5 | Right middle occipital sulcus and lunatus sulcus | 0.244 |

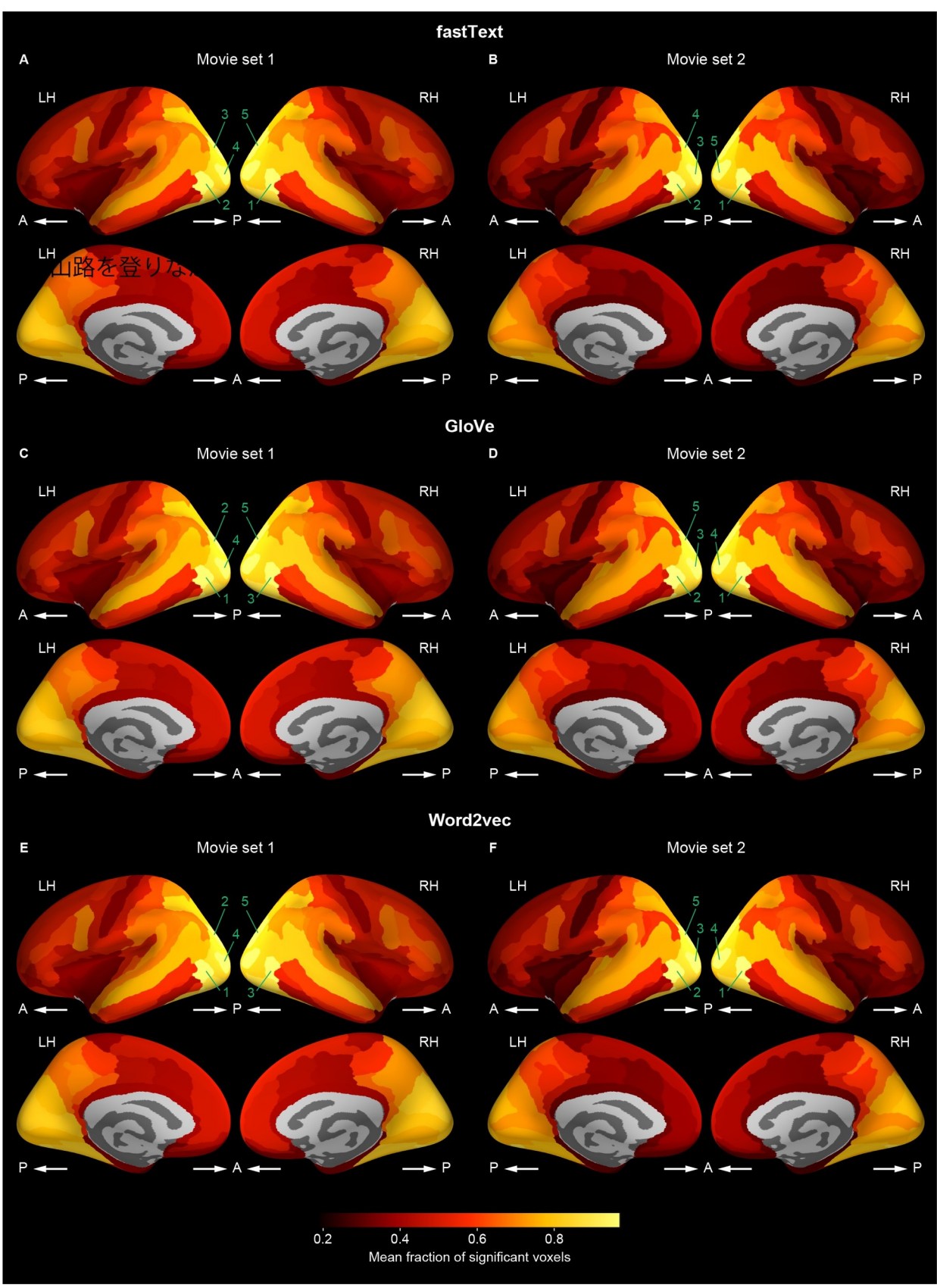

**Fig 5. Cortical mapping of fraction of significant voxels.** Participant-averaged fraction of significant voxels of voxelwise models (vector dimension = 1000) mapped onto the cortical surface of a reference brain for each of fastText (**A** and **B**), GloVe (**C** and **D**), and word2vec (**E** and **F**) vectors and for each of movie sets 1 (**A**, **C**, and **E**) and 2 (**B**, **D**, and **F**). The fraction was computed within each cortical region. Brighter colors indicate regions that have larger fraction. The five cortical regions with the highest mean fraction of significant voxels are numbered in a descending order separately for each type of word vectors and for each movie set. The names of these regions are shown in Table 2. Other conventions are the same as in Fig 4.

In comparison between word vectors and discrete word features, we observed similar localization patterns of prediction accuracy (S15 Fig and S13 Table) and the fraction of significant voxels (S16 Fig and S14 Table). Therefore, even regarding the localization of predictable cortical regions, we did not find any clear differences between word vectors and discrete word features.

To assess the effectiveness of the semantic relational structure of word vectors in response prediction in each cortical region, we compared prediction performance (i.e., prediction performance and the fraction of significant voxels) between the voxelwise model based on trained word vectors and that based on untrained word vectors within each region. The results reveal

**Table 2. Cortical regions with the highest mean fraction of significant voxels for each type of word vectors and for each movie set.**

| | | Rank | Region name | Fraction of significant voxels |
|---|---|---|---|---|
| **fastText** | **Movie set 1** | 1 | Right anterior occipital sulcus and preoccipital notch | 0.946 |
| | | 2 | Left anterior occipital sulcus and preoccipital notch | 0.945 |
| | | 3 | Left superior occipital sulcus and transverse occipital sulcus | 0.943 |
| | | 4 | Left middle occipital sulcus and lunatus sulcus | 0.935 |
| | | 5 | Right superior occipital sulcus and transverse occipital sulcus | 0.928 |
| | **Movie set 2** | 1 | Right anterior occipital sulcus and preoccipital notch | 0.966 |
| | | 2 | Left anterior occipital sulcus and preoccipital notch | 0.962 |
| | | 3 | Left middle occipital sulcus and lunatus sulcus | 0.943 |
| | | 4 | Right middle occipital sulcus and lunatus sulcus | 0.937 |
| | | 5 | Left superior occipital sulcus and transverse occipital sulcus | 0.930 |
| **GloVe** | **Movie set 1** | 1 | Left anterior occipital sulcus and preoccipital notch | 0.953 |
| | | 2 | Left superior occipital sulcus and transverse occipital sulcus | 0.950 |
| | | 3 | Right anterior occipital sulcus and preoccipital notch | 0.948 |
| | | 4 | Left middle occipital sulcus and lunatus sulcus | 0.938 |
| | | 5 | Right superior occipital sulcus and transverse occipital sulcus | 0.937 |
| | **Movie set 2** | 1 | Right anterior occipital sulcus and preoccipital notch | 0.969 |
| | | 2 | Left anterior occipital sulcus and preoccipital notch | 0.968 |
| | | 3 | Left middle occipital sulcus and lunatus sulcus | 0.946 |
| | | 4 | Right middle occipital sulcus and lunatus sulcus | 0.939 |
| | | 5 | Left superior occipital sulcus and transverse occipital sulcus | 0.935 |
| **Word2vec** | **Movie set 1** | 1 | Left anterior occipital sulcus and preoccipital notch | 0.952 |
| | | 2 | Left superior occipital sulcus and transverse occipital sulcus | 0.950 |
| | | 3 | Right anterior occipital sulcus and preoccipital notch | 0.948 |
| | | 4 | Left middle occipital sulcus and lunatus sulcus | 0.940 |
| | | 5 | Right superior occipital sulcus and transverse occipital sulcus | 0.937 |
| | **Movie set 2** | 1 | Right anterior occipital sulcus and preoccipital notch | 0.969 |
| | | 2 | Left anterior occipital sulcus and preoccipital notch | 0.965 |
| | | 3 | Left middle occipital sulcus and lunatus sulcus | 0.948 |
| | | 4 | Right middle occipital sulcus and lunatus sulcus | 0.941 |
| | | 5 | Left superior occipital sulcus and transverse occipital sulcus | 0.934 |

that the model based on trained vectors exhibited significantly higher response-prediction performance in the majority of the 148 regions (S17 Fig; 130–144 regions for prediction accuracy; 136–148 regions for the fraction of significant voxels; Wilcoxon test, p < 0.05, FDR corrected). Even across regions, the model based on trained vectors outperformed the model based on untrained vectors regardless of performance measures, word-vector types, and datasets (Wilcoxon test, p < 0.05, FDR corrected). Although performance differences between these two models were not large, the improvement of the performance was observed comparably across regions rather than conspicuously in specific regions. These results suggest that the semantic relational structure of words, captured by word vectors, improves the modeling of semantic representations in widespread brain regions.

## Correlation between brain- and behavior-derived word dissimilarities

Next, we tested the correlation between the word dissimilarity matrix derived from the voxelwise models and that derived from behavioral data to clarify the behavioral correlates of modeled semantic representations. Word representations in the modeled brain space were calculated by multiplying the original fastText, GloVe, or word2vec word vectors by model weights, and the brain-derived word dissimilarity matrix was obtained from the correlation distance between all possible pairs of word representations (Fig 1B). Meanwhile, the behavior-derived word dissimilarity matrix was measured from the behavioral data from the word-arrangement task (Fig 2), in which 36 participants completed 18.8 (SD = 10.7) trials on average for each session (see Methods for more details). These brain- and behavior-derived word dissimilarity matrices were constructed by averaging the matrices over all brain models or all behavioral data separately for nouns and adjectives. In addition, to determine whether the behavioral correlates of word dissimilarity change through the transformation from original word vector representations to brain representations, we also calculated word vector-derived word dissimilarity matrices from the correlation distance between all possible pairs of original fastText, GloVe, or word2vec word vectors separately for nouns and adjectives.

Fig 6 shows the behavior-derived word dissimilarity matrices and the brain- and word vector-derived word dissimilarity matrices constructed using 1000-dimensional fastText word vectors. The brain-derived matrices were highly consistent across movie sets (Spearman's ρ = 0.926 and 0.943 for nouns and adjectives, respectively). For both nouns (Fig 6A) and adjectives (Fig 6B), we found significant correlations between the brain- and behavior-derived matrices (brain–behavior correlations; permutation test, p < 0.0001). The correlation coefficients were larger for nouns than for adjectives (p < 0.0001). Although we also found significant correlations between the word vector- and behavior-derived matrices (word vector–behavior correlations; p < 0.0001), the brain–behavior correlations for nouns (but not for adjectives) were significantly stronger than the word vector–behavior correlations (p < 0.0001).

Fig 7 summarizes the brain–and word vector–behavior correlations for each word-vector type (i.e., fastText, GloVe, or word2vec; vector dimensionality = 1000). For all the vector types and for both nouns and adjectives, there were significant brain–behavior correlations as well as word vector–behavior correlations (permutation test, p < 0.0001, FDR corrected). Such significant correlations were observed regardless of vector dimensionality (S18–S20 Figs); although the correlation coefficients differed across vector dimensions, no clear tendency of changes in correlation coefficients across vector dimensions was determined. These results indicate that the brain-derived dissimilarity structure of semantic representations correlates with the behavior-derived dissimilarity structure regardless of vector types and dimensionality.

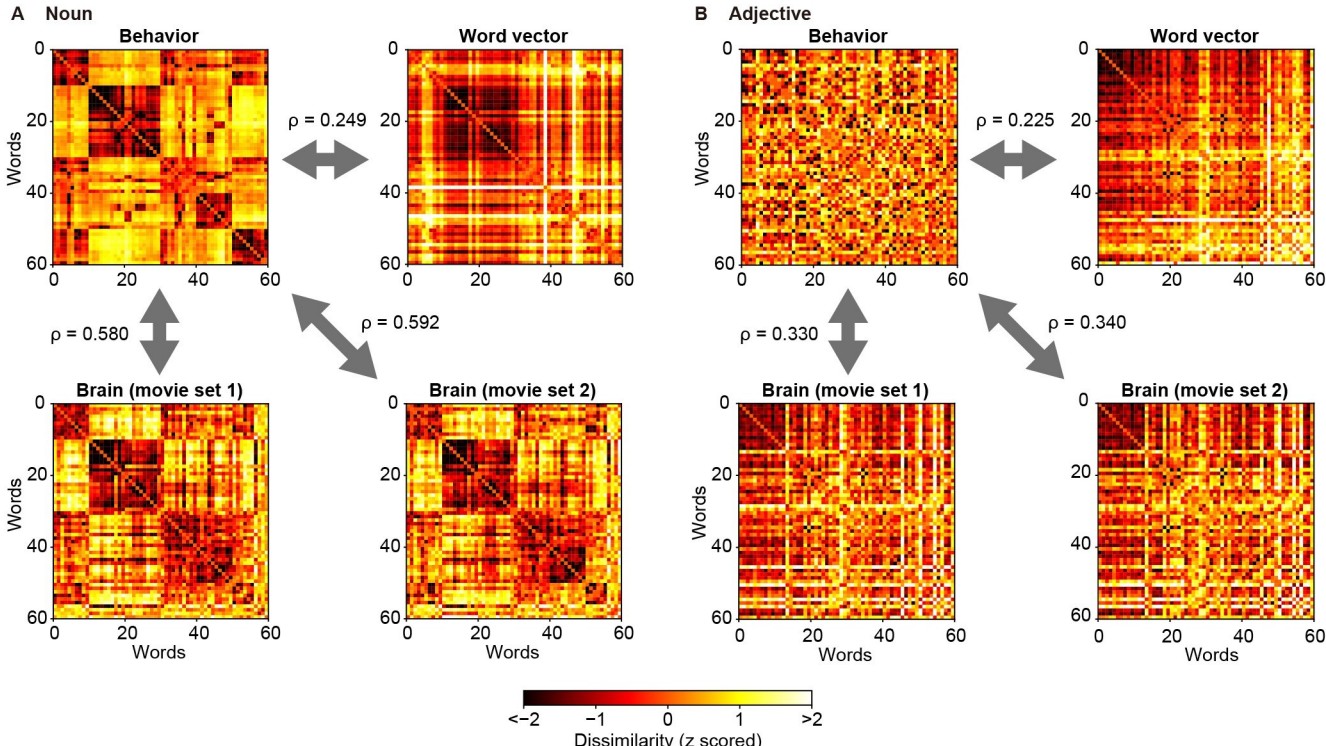

**Fig 6. Correlations between example dissimilarity matrices.** We have constructed word dissimilarity matrices separately for nouns (**A**) and adjectives (**B**). Each color map shows the behavior-derived matrices obtained from behavioral data (top left in **A** and **B**), the word vector-derived matrices obtained directly from 1000-dimensional fastText vectors (top right), or the brain-derived matrices obtained from voxelwise models (based on 1000-dimensional fastText vectors) for each movie set (bottom). Brighter colors in each map indicate higher dissimilarity of word pairs. Spearman's correlation coefficients (ρ) are indicated.

We then performed three different comparisons of behavioral correlations in Fig 7. In the comparison between nouns and adjectives, the brain–behavior correlations of noun dissimilarity were significantly stronger than those of adjective dissimilarity for all the vector types (permutation test, p < 0.0001, FDR corrected). In the comparison between brain–behavior and word vector–behavior correlations, there was a significant difference only for fastText noun dissimilarity (p < 0.0001, FDR corrected). In the comparison between vector types, the vector-type difference of brain–behavior correlations was partly significant but small (p < 0.05, FDR corrected) whereas the word vector–behavior correlations for fastText vectors was much weaker than those for the other types of vectors (p < 0.0001, FDR corrected).

In these cases, the populations from which we obtained these dissimilarity matrices were different between brain- and behavior-derived data. However, we observed a significant correlation for both nouns and adjectives and for both movie sets even when these matrices were collected from and averaged over the same population (six participants; permutation test, p < 0.0001; S21 Fig). At the level of individual participants, the brain–behavior correlation of noun dissimilarity was significant for each of all the six participants from this population (p < 0.01) whereas the correlation of adjective dissimilarity was significantly higher than chance level for only three participants of them (p < 0.05; S22 Fig). Together, the word vector-based models could capture perceptual noun and adjective dissimilarities at the population level and perceptual noun dissimilarity at the individual level even when the brain- and behavior-derived word dissimilarities were originated from the same population.

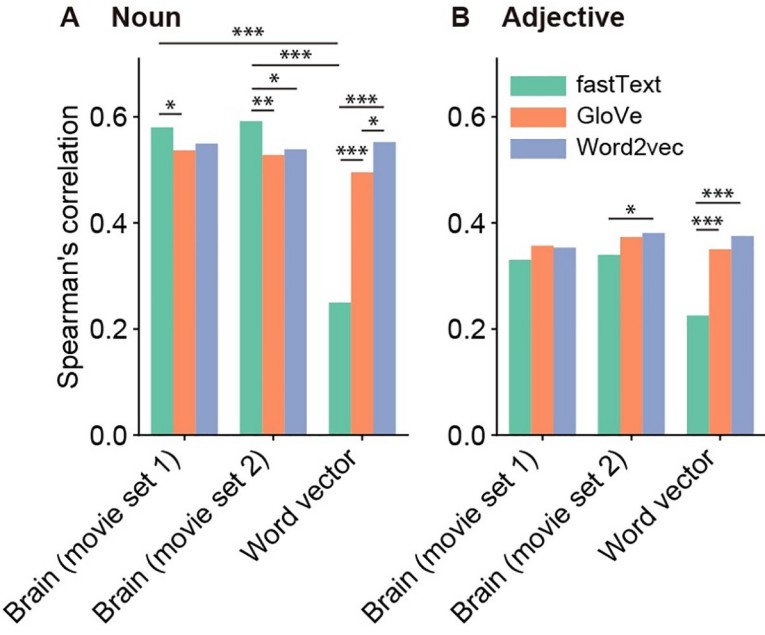

**Fig 7. Behavioral correlates of brain-derived word dissimilarity for different word vector types.** We computed brain–behavior and word vector–behavior correlations separately for the three types of 1000-dimensional word vectors (green, fastText; orange, GloVe; blue, word2vec). The Spearman's correlation coefficients are separately shown for nouns (**A**) and adjectives (**B**). Marks above bars indicate the statistical significance of the correlation difference between different word vector types and between brain–behavior and word vector–behavior correlations (permutation test, ***p < 0.0001, **p < 0.01, *p < 0.05, FDR corrected).

The brain-derived dissimilarity matrices described above (Figs 6 and 7 and S18–S22) were calculated from the model weights of all cortical voxels (the number of voxels was 54914–73301 [mean ± SD = 62245 ± 5018] for movie set 1 and 52832–73301 [mean ± SD = 62150 ± 5190] for movie set 2). However, the high performance of the voxelwise models was observed in localized cortical regions (Figs 4, 5, S6–S8 and S11–S13). Hence, the correlation of brain- and behavior-derived dissimilarity matrices may be stronger when only the cortical regions with high model performance were used. To test this possibility, we constructed brain-derived dissimilarity matrices using only those voxels with the highest prediction accuracy (top 2000, 5000, 10000, 30000, or 50000 voxels). Then, their correlations with behavior-derived matrices were compared with the original correlations calculated using all cortical voxels. However, we found the strongest correlation when using all cortical voxels regardless of word vector types (Figs 8 and S23–S25). This result suggests that semantic information correlated with behavior are distributed broadly across the cortex and can be captured using word vector-based voxelwise models.

To gain a further understanding of cortical localization of behaviorally correlated semantic representations modeled by word vectors, we examined brain–behavior correlations of word dissimilarity within each cortical region. We found that although moderate or strong correlations were observed across the cortex (Spearman's ρ = 0.356–0.621 for noun dissimilarity, 0.246–0.400 for adjective dissimilarity), the correlations were relatively weak in low-level audiovisual regions, such as the posterior occipital cortex and the superior temporal area (Figs 9 and 10). This localization pattern of brain–behavior correlations across the cortex was deemed different from that of model prediction performance (Figs 4, 5, S6–S8 and S11–S13). This discrepancy is likely to cause the reduced brain–behavior correlation when voxels used for the analysis were selected according to high prediction accuracy (Figs 8 and S23–S25).

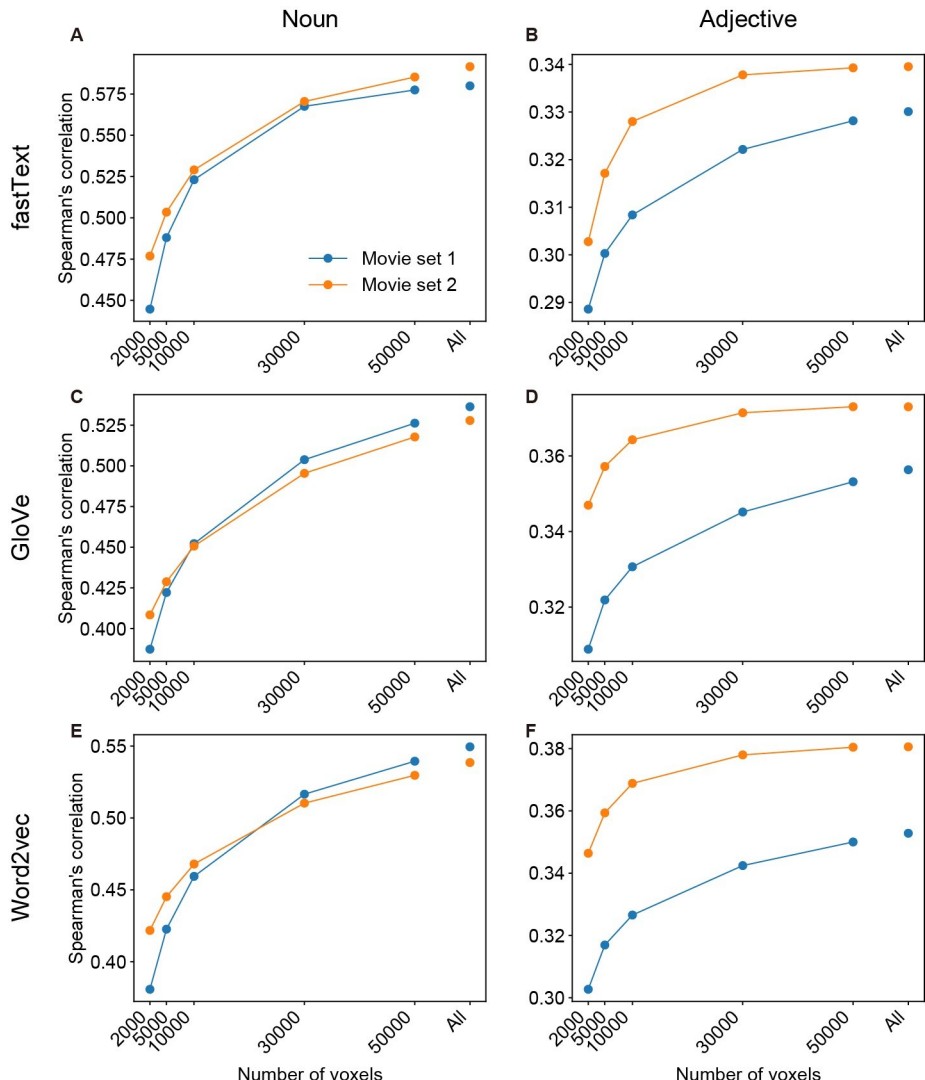

**Fig 8. Effects of voxel selection on brain–behavior correlation.** We have evaluated the correlations between brain- and behavior-derived word dissimilarity matrices when the brain-derived matrix was obtained from the selected voxels with the highest model accuracy. The correlation coefficients for fastText (**A** and **B**), GloVe (**C** and **D**), and word2vec (**E** and **F**) vectors are shown separately for nouns (**A**, **C**, and **E**) and adjectives (**B**, **D**, and **F**), while the numbers of selected voxels are changed (2000, 5000, 10000, 30000, 50000, and all voxels).

Considering that the fraction of significantly predictable voxels was sufficiently large even in the high-level cortical regions showing strong brain–behavior correlations (Fig 5 and S11– S13), these results suggest that word vector-based models effectively capture semantic information in these regions, which is more closely associated with human perception than the low- level audiovisual regions.

Previous studies have reported that cortical representations in the object-selective visual areas show clear distinction between animate and inanimate categories [27–29]. Since the nouns we used for constructing word dissimilarity matrices consisted of six different semantic categories along animate/inanimate dimension (humans, non-human animals, non-animal nature things, constructs, vehicles, and other artifacts; see also Methods), it might be argued that the observed behavioral correlates of brain-derived word dissimilarity simply reflected

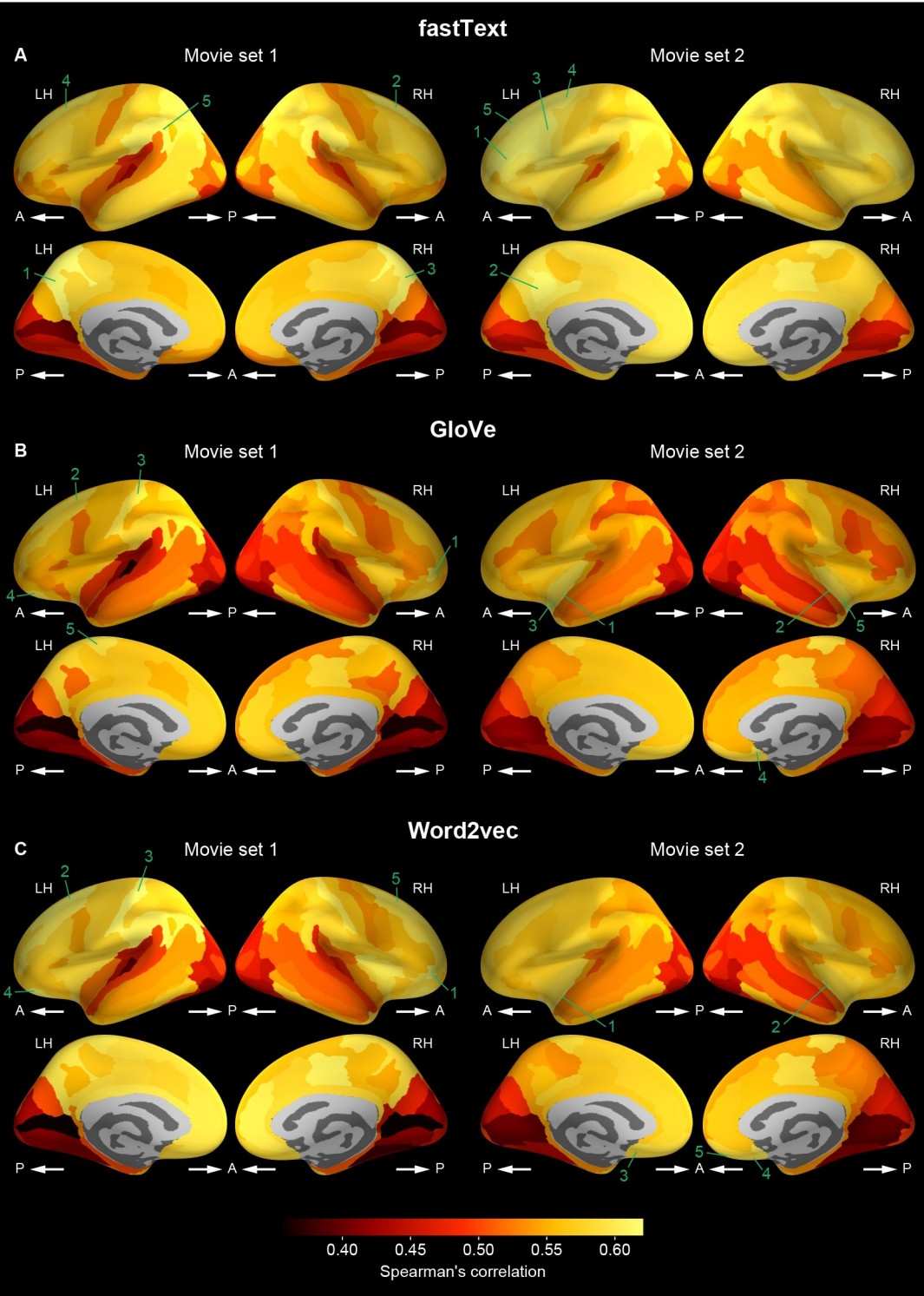

**Fig 9. Cortical mapping of brain–behavior correlations of noun dissimilarity.** Spearman's correlation coefficients between brain- and behavior-derived noun dissimilarity matrices were calculated within each cortical region and mapped onto the cortical surface of a reference brain for each of fastText (**A**), GloVe (**B**), and word2vec (**C**) models (vector dimension = 1000) and for each of movie sets 1(left) and 2 (right). Brighter colors indicate regions that have larger correlation coefficients. The five cortical regions with the highest correlation coefficients are numbered in a descending order separately for each type of word vectors and for each movie set. The names of these regions are shown in S15 Table. Other conventions are the same as in Fig 4.

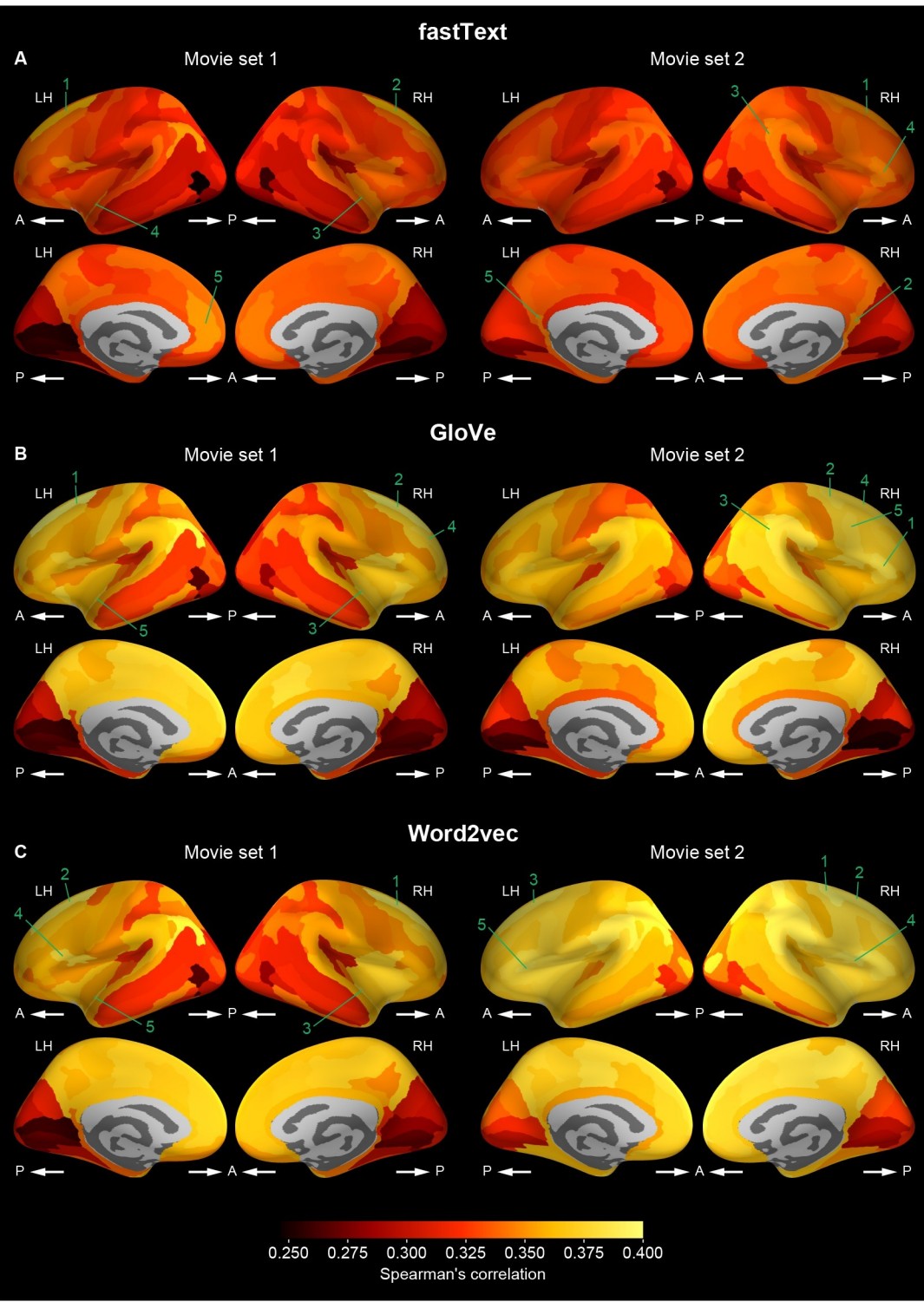

**Fig 10. Cortical mapping of brain–behavior correlations of adjective dissimilarity.** The same analysis as in Fig 9 but for adjective dissimilarity. The names of the five cortical regions with the highest Spearman's correlation coefficients for each word-vector type and each movie set are shown in S16 Table.

such clear distinction of animate and inanimate representations. However, even when brain–behavior correlations were examined separately for each of the six categories, we found significant brain–behavior correlations for each of the "humans", "non-animal natural things", "vehicles", and "other artifacts" categories (p < 0.05, FDR corrected; S26 Fig). This result indicates that the observed brain–behavior correlations of word dissimilarity cannot be explained only by simple representational distinction between animate and inanimate categories.

Finally, we compared the brain-derived word dissimilarity modeled by word vectors and that modeled by discrete word features in terms of their behavioral correlates. In contrast to the performance of brain-response prediction (S4, S15 and S16 Figs), the brain–behavior correlations of word dissimilarity for the binary-labeling models were much weaker than those for the word vector-based models regardless of datasets and vector dimensionality (S27 Fig). This result suggests that the voxelwise modeling based on word vectors have a clear advantage in capturing the similarity structure of cortical semantic representations associated with human perception.

## Discussion

We have examined the behavioral correlates of semantic representations estimated by word vector-based brain models. We constructed a voxelwise model that has the ability to predict movie-evoked fMRI signals in individual brains through a word vector space. The voxelwise models showed substantial response-prediction performance in reasonable cortical regions consistently across stimulus and parameter sets. There were significant correlations between the word dissimilarity structure derived from the voxelwise models and that derived from behavioral data. These results suggest that the semantic representations estimated from word vector-based brain models can appropriately capture the semantic relationship of words in human perception. Our findings contribute to the establishment of word vector-based brain modeling as a powerful tool in investigating human semantic processing.

Word vectors have been extensively leveraged for the modeling of semantic information in the human brain. Some studies have successfully predicted brain responses, through word vector spaces, evoked by audiovisual stimuli, such as movies [15,16,22], pictures [10,13], and sentences [14]. In addition, one of these studies has visualized the semantic representational space in the brain through word vector-based modeling [16]. Another line of studies has successfully recovered perceived semantic contents from brain responses by using word vector spaces [11,17,18]. In this way, word vectors will be a useful tool to investigate comprehensively the semantic processing in the human brain. Several studies have reported the association between original word vectors and human behavior [6,30–32]. However, no study has explicitly clarified the behavioral correlates of the brain semantic representations revealed by word vector-based models. Although previous studies in which movie-evoked brain responses were predicted from manual descriptions for the movie scenes [11,15] employed human behavior implicitly, they did not reveal the association between brain- and behavior-derived semantic representations. To the best of our knowledge, this is the first study demonstrating the behavioral correlates of semantic representations modeled by word vectors.

We demonstrated that word vector-based voxelwise models much outperformed voxelwise models with discrete word features in terms of the brain–behavior correlation of word dissimilarity (S27 Fig) although response-prediction performance and its cortical localization were deemed comparable between these models (S4, S15 and S16 Figs). This finding suggests that word vectors are more effective to model the similarity structure of cortical semantic representations associated with human perception, compared with discrete word features, which were used for voxelwise modeling in previous studies (e.g., [23]). In addition, another advantage of

word vector-based modeling is that semantic representations can be quantified using much more vocabularies [11] than modeling based on discrete word features. This enables to comprehensively understand the representational structure of semantic information in the human brain. For example, a recent study demonstrated the multidimensional representations of natural objects on the basis of a data-driven approach using large-scale behavioral data of similarity judgements [33]. The word vector-based models capturing human word similarity judgements may be used to explore whether such a multidimensional structure of semantic representations actually exists in the human brain.

In addition, the word vector-based modeling with large vocabularies allows the quantification of different aspects of semantic information in the brain by using different types of words. In particular, adjectives have a potential for the visualization of impression information [11]. We found the behavioral correlates of modeled semantic representations on nouns and even adjectives, which were consistently observed across stimulus and parameter sets (Figs 7, S18–S20). These findings indicate that word vectors are deemed suitable for modeling the cortical representations of impression information. Thus, word vector-based modeling provides a useful framework for extensive investigation of semantic representations in the brain.

In this study, we found that the cortical regions showing high accuracy in predicting movie-evoked brain response by our models were localized primarily in the posterior and temporal cortices (Figs 4 and S6–S8). This localization pattern of high predictability is consistent with previous reports in which cortical semantic representations were modeled from movie-evoked response [12,23]. However, when semantic models using word features are applied to brain response evoked by linguistic stimuli, e.g., sentences and speech, the brain response is highly predictable not only in sensory regions but also in the language network, including the left inferior frontal and superior temporal cortices [14,21,34]. These findings imply that the localization of high predictability depends on stimulus modalities, namely, whether stimuli are linguistic or not. This notion is supported by evidence that the language network has a relatively small contribution to semantic processing that requires no explicit linguistic processing [35]. Thus, the semantic representations modeled using the brain response evoked by nonlinguistic stimuli may be biased toward specific modalities, such as vision and audition.

However, the observed localization pattern of high predictability does not mean that our models fail to capture semantic information in high-level cortical regions, including the language network. We also found that the fraction of voxels being significantly predictable by our models was sufficiently high in widespread cortical regions (Figs 5 and S11–S14). In addition, significant brain–behavior correlations of word dissimilarity were observed broadly across the cortex and rather stronger in the high-level association cortex than in the low-level audiovisual cortex (Figs 9 and 10). In general, the neural signals in high-level cortical regions explain a larger portion of the variability in human perception and behavior than those in low-level cortical regions [36–38]. Thus, the results suggest that our word vector-based models can affectively capture semantic information in high-level cortical regions which is closely associated with human perception.

In the field of natural language processing, recently developed algorithms based on deep learning have exhibited state-of-the-art performance in many types of natural language tasks [39–41]. The feature representations obtained by these algorithms are also beginning to be used for the modeling of semantic representations [42,43], which show state-of-the-art performance even in brain-response prediction [44]. Hence, it is possible that the correlation between brain- and behavior-derived semantic representations improves by using such deep learning features instead of word vectors. However, some advantages are noted using word vectors compared with deep learning features: First, word vectors are easily trained with much smaller sets of parameters than deep learning features. Second, word vectors are easily

interpretable owing to their direct association with the meaning of words. In contrast, deep learning features cannot be easily interpreted without sophisticated methods for improving their interpretability [45,46]. Third, there are many existing methods to improve word-vector spaces for specific types of tasks [13,47–51]. These advantages may be also beneficial for the modeling of semantic representations. Therefore, it is important to use these two types of algorithms separately depending on the aims of specific studies on semantic processing.

## Methods

### Ethics statement

The experimental protocol was approved by the ethics and safety committees of the National Institute of Information and Communications Technology. Written informed consent was obtained from all of experimental participants.

### Participants

In total, 52 healthy Japanese participants (21 females and 31 males; age 20–61, mean ± SD = 26.8 ± 8.9 years) were recruited for the 2 sets of fMRI experiments. Among the recruits, 36 participated in one or the other of these fMRI experiments, and 16 participated in both. In addition, another 36 healthy Japanese participants (20 females and 16 males; age 18–58, mean ± SD = 25.9 ± 10.1 years) were recruited for psychological experiments. Of these, 6 were also participants of the fMRI experiments and 30 were unique participants. All participants had normal or corrected-to-normal vision. The fMRI data, but not the psychological data, used here were also utilized in a previous publication [22].

### MRI experiments

Functional and anatomical MRI data were collected via a 3T Siemens MAGNETOM Prisma scanner (Siemens, Germany), with a 64-channel Siemens volume coil. Functional data were collected using a multiband gradient echo EPI sequence [52] (TR = 1000 ms; TE = 30 ms; flip angle = 60˚; voxel size = 2 × 2 × 2 mm; matrix size = 96 × 96; FOV = 192 × 192 mm; the number of slices = 72; multiband factor = 6). Anatomical data were also gathered using a T1-weighted MPRAGE sequence (TR = 2530 ms; TE = 3.26 ms; flip angle = 9˚; voxel size = 1 × 1 × 1 mm; matrix size = 256 × 256; FOV = 256 × 256 mm; the number of slices = 208) on the same 3T scanner.

In the two sets of fMRI experiments, participants were asked to view movie stimuli on a projector screen inside the scanner (27.9 × 15.5 of visual angle at 30Hz) and used MR-compatible headphones for the sounds. The participants were given no explicit task. The fMRI data for each participant upon viewing of the movies were collected in three separate recording sessions over 3 days for each set of fMRI experiments.

The movie stimuli consisted of Japanese television advertisements for one set of experiments (movie set 1) and Japanese web advertisements for the other set of experiments (movie set 2; see also [22]). Movie set 1 has included 420 ads broadcasted on Japanese TV between 2011 and 2017; meanwhile, movie set 2 included 368 ads broadcasted on the Internet between 2015 and 2018. The ad movies were all unique, include a wide variety of product categories (see S17 Table for more details). The length of each movie was 15 or 30 s. To create the movie stimuli for each experiment, the original movies in each of the movie sets 1 and 2 were sequentially concatenated in a pseudo-random order. For each movie set, 14 non-overlapping movie clips of 610 s in length were obtained.

Individual movie clips were then displayed in separate scans. The initial 10 s of each clip served as a dummy in order to discard hemodynamic transient signals caused by clip onset. fMRI responses collected during the 10 s dummy part were not used for modeling. Twelve clips from each movie set were only presented once. The fMRI responses to these clips were used for the training of the voxelwise models (training dataset; 7200 s in total). The other two clips for each movie set were presented four times each in four separate scans. The fMRI responses to these clips were then averaged across four scans to improve the signal-to-noise ratio [11,23,53]. The averaged responses were further used for the test of the voxelwise models (test dataset; 1200 s in total).

## MRI data pre-processing

Motion correction in each functional scan was carried out via the statistical parameter mapping toolbox (SPM8, http://www.fil.ion.ucl.ac.uk/spm/software/spm8/). For each subject, all volumes were aligned to the first image from the first functional run. Low-frequency fMRI response drift was then eliminated by subtracting median-filtered signals (within a 120-s window) from raw signals. Then, the response for each voxel was normalized by subtracting the mean response and scaling to the unit variance. FreeSurfer [24,25] was used to identify cortical surfaces from anatomical data and register these surfaces to the voxels of functional data. Then, each voxel was assigned to one of the 148 cortical regions derived from the Destrieux atlas for cortical segmentation [26].

## Word vector spaces

This study used fastText skip-gram [4], GloVe [3], or word2vec skip-gram [2] to construct a word vector space. These algorithms have been originally developed to learn a word vector space based on word co-occurrence statistics in natural language texts.

The training objective of the skip-gram algorithm of fastText and word2vec is to obtain word vector representations that enable the surrounding words to be accurately predicted from a given word in a sentence [2,4]. More formally, given a sequence of training words $w_1$, $w_2$,. . .,$w_T$, the skip-gram algorithm seeks a $K$-dimensional vector space that maximizes the average log probability, given as:

$$\frac{1}{T}\sum_{t=1}^{T}\sum_{-c\leq j\leq c, j\neq 0}\log p(w_{t+j}|w_t)$$

where $c$ is the size of the context window, which corresponds to the number of to-be-predicted words before and after the center word $w_t$. Therefore, the skip-gram vector space is optimized on the basis of the local co-occurrence statistics of nearby words in the text corpus. The basic formulation of $p(w_{t+j}|w_t)$ is the softmax function:

$$p\left(w_y|w_x\right) = \frac{\exp(\langle \mathbf{v}_{wx}, \mathbf{v}_{wy}\rangle)}{\sum_{i=0}^{W}(\langle \mathbf{v}_{wx}, \mathbf{v}_{wi}\rangle)}$$

where $\mathbf{v}_{wi}$ is the vector representation of $w_i$, $W$ is the number of words in the vocabulary, and $\langle \mathbf{v}_1, \mathbf{v}_2\rangle$ indicates the inner product of vectors $\mathbf{v}_1$ and $\mathbf{v}_2$. However, because of the high computational cost of this formulation, the negative sampling technique is used to produce a computationally efficient approximation of the softmax function [2]. In addition, fastText introduces sub-word modeling, which is robust for inflected and rare words [4].

GloVe learns word vector representations using a global log-bilinear regression model that is subject to the cost function:

$$J = \sum_{i,j=1}^{V} f(X_{ij})(\mathbf{w}_i^T \tilde{\mathbf{w}}_j + b_i + \tilde{b}_j - \log X_{ij})^2$$

where $V$ is the vocabulary size, $\mathbf{w}_i \in \mathbb{R}^d$ is a $d$-dimensional vector of word $i$, $\tilde{\mathbf{w}}_j \in \mathbb{R}^d$ is a separate context vector of word $j$, and $b_i$ and $\tilde{b}_j$ are additional biases. $X$ is the co-occurrence matrix, where $X_{ij}$ represents the number of occurrences of word $i$ in the context of word $j$. $f(X_{ij})$ is a specific weighting function. GloVe combines the advantages of local context window and global matrix factorization methods [3].

A vector space of each algorithm was constructed from a text corpus of the Japanese Wikipedia dump on April 1, 2020. All Japanese texts in the corpus were segmented into words and lemmatized using MeCab (http://taku910.github.io/mecab). Used were only nouns, verbs, and adjectives. In an attempt to improve the reliability of word-vector learning, the vocabulary size was restricted to ~100,000 words by excluding words that appeared infrequently in the corpus. The learning parameters of fastText and word2vec used were as follows: window size = 10; the number of negative samples = 5; downsampling rate = 0.001; and the number of learning epochs = 10. The learning parameters of GloVe used were as follows: window size = 10; initial learning rate = 0.05; $\alpha$ = 0.75; and the number of training iterations = 50. The vector spaces were constructed using five different numbers of vector dimensions (100, 300, 500, 1000, and 2000). Of these, the 1000-dimension vector space of each algorithm was used for the main analysis (Figs 3–10), and the other dimensions were used to test the effect of vector dimensionality on modeling performance and behavioral correlates. To eliminate the effects of trial variations in the learning quality of vector spaces of each algorithm on the performance of voxelwise modeling, five different vector spaces were learned independently using the same text corpus and the same algorithm with the same parameters. All results were obtained from the average over five learned spaces for each parameter set.

## Movie scene descriptions

Manual scene descriptions using natural Japanese language were provided for every 1-s scene of each movie in movie sets 1 and 2, in a manner similar to that described previously [11,12,22]. The annotators were native Japanese speakers (movie set 1: 68 females and 28 males, age 19–62 years; movie set 2: 11 females and 2 males, age 20–56 years), who were not the fMRI participants. They were instructed to describe each scene (the middle frame of each 1-s clip) using more than 50 Japanese characters. Multiple annotators (movie set 1, 12–14 annotators; movie set 2, 5 annotators) were randomly assigned for each scene to reduce the potential effect of personal bias. The descriptions contain a variety of expressions reflecting not only objective perceptions but also the subjective perceptions of the annotator (e.g., impression, feeling, association with ideas; for more details, see https://osf.io/3hkwd).

Each description for a given scene was also segmented, lemmatized, and decomposed into nouns, verbs, and adjectives via MeCab as described above; then, they were transformed into fastText, GloVe, or word2vec vectors. The word vectors were then averaged within each description. For each scene, all vectors obtained from the different descriptions were averaged. Through this procedure, a single vector (scene vector) was obtained for each 1-s scene, which was later used for modeling.

### Binary labeling of movie scenes

To compare discrete word features with word vectors in terms of the modeling of brain semantic representations, the binary labeling of semantic contents in each movie scene was used to construct voxelwise models. The binary labels were obtained from the presence/absence of words in the manual movie-scene descriptions. For each scene, individual words (nouns, verbs, and adjectives) in the vocabulary were counted in all descriptions for the scene. If a given word was present, the binary label of the word was 1. If not, the binary label of the word was 0. This labeling was performed over all movie scenes. Then, scene vectors with the dimensionality of 100, 300, 500, 1000, and 2000 were obtained from the binary labels of 100, 300, 500, 1000, and 2000 words that most frequently appeared in the movie-scene descriptions for each movie set. This type of semantic features is similar to ones used in a previous study on the modeling of cortical semantic representations [23]. However, the previous study performed the binary labeling of movie scenes using the manual labeling of single words by one annotator, unlike our binary labeling.

### Voxelwise modeling

The procedure of voxelwise modeling was similar to that described previously [12]. A series of fMRI responses evoked in individual $N$ voxel by a series of $S$ movie scenes, represented by a $S \times N$ matrix $\mathbf{R}$, was modeled as a weighted linear combination of a $K$-dimensional scene-vector matrix $\mathbf{V}$ plus isotropic Gaussian noise $\boldsymbol{\varepsilon}$.

$$\mathbf{R} = \mathbf{V}\mathbf{W} + \boldsymbol{\varepsilon}$$

A set of linear temporal filters was used to model the slow hemodynamic response and its coupling with brain response [53]. In an attempt to capture the hemodynamic delay in the responses, the $S \times 4K$ scene-vector matrix was constructed by concatenating four sets of $K$-dimensional scene vectors with delays of 3, 4, 5, and 6 s. The $4K \times N$ weighted matrix $\mathbf{W}$ was estimated using an L2-regularized linear least-squares regression, which can obtain good estimates even for models containing a large number of regressors [23].

The model training was conducted using the training dataset (7200 datapoints). In order to estimate the regularization parameters, the training dataset was randomly divided into two subsets containing 80% and 20% of the samples, for model fitting and validation, respectively. This random resampling procedure was repeated 10 times. Regularization parameters were optimized, according to the mean Pearson's correlation coefficient between the predicted and measured fMRI signals for the 20% validation samples. An optimal parameter was obtained separately for each model.

The brain-response prediction performance of the model was evaluated using the test dataset (1200 datapoints), which was not used for model fitting or parameter estimation. The prediction accuracy of the model was quantified as the Pearson's correlation coefficient between the predicted and average measured fMRI signals in the test dataset. In addition, the significance of prediction accuracy (Pearson's correlation) for each of all cortical voxels was evaluated after the correction for multiple comparisons using FDR. The rate of voxels, to all cortical voxels, for which the prediction accuracy reached a significance threshold ($p < 0.05$, FDR corrected) was calculated as the fraction of significant voxels, which was another measure of brain-response prediction performance for each model.

To test whether the fraction of significant voxels for each cortical region was significantly above chance level, two types of statistical tests were performed. First, using the fraction for each model of each participant as a data sample, whether or not the fraction was significantly higher than a standard significance threshold ($p = 0.05$) was tested using Wilcoxon test with

FDR correction. Second, the chance level of the fraction for each cortical region was more conservatively estimated using control voxelwise models based on word vectors that were shuffled across vector dimensions for each vector. Five different sets of the shuffled word vectors were generated from each word-vector space and used to construct five different control voxelwise models for each participant. Then, the fraction of significant voxels was obtained for each cortical region using these control models and averaged across the five different models, which produces the chance level of the fraction for each participant. Whether the fraction of significant voxels for each original model was significantly above the chance level was tested for each region using Wilcoxon test with FDR correction while the fraction for each participant was used as a data sample.

Word vectors effectively capture the semantic relational structure of words as demonstrated by previous studies that applied word vectors to word similarity judgements [3,4,6]. To determine whether such semantic relational structure of word vectors is effective in the modeling of cortical semantic representations, the response-prediction performance of voxelwise models using original word vectors was compared with that using untrained word vectors. Untrained word vectors were obtained with the same procedure as used for the generation of original, trained word vectors, except that the statistical training of word vectors was omitted. This procedure yielded 1000-dimensional random vectors that were each assigned to each word in the vocabulary shared with the corresponding trained vectors. Hence, untrained vectors had signatures of individual words but not the semantic relational structure of words at all. Then, scene vectors were generated from untrained vector representations of scene descriptions.

Voxelwise models based on untrained vectors were constructed using the same procedure as used for voxelwise models based on trained vectors. Five different voxelwise models for each participant were constructed from five different sets of untrained vectors. The performance of brain-response prediction (i.e., prediction accuracy and the fraction of significant voxels) for each participant was obtained by averaging the performance of these five models. Finally, the performance for the models based on untrained vectors was statistically compared with that for the models based on trained vectors. Higher performance for the models based on trained vectors indicates that the semantic relational structure of word vectors is effective in the modeling of cortical semantic representations.

## Psychological experiments

Participants were asked to perform a word-arrangement task on a PC. This task is considered to be a modified version of the psychological task introduced previously [19]. Kriegeskorte and colleagues originally used this task to examine the behavioral correlates of cortical object representations, which were quantified using the representational similarity of fMRI signals evoked by object stimuli [54,55]. The present study used this task to study the methodological validity of word vector-based brain modeling by testing the behavioral correlates of modeled cortical semantic representations. The psychological data are available online (https://osf.io/um3qg/).

In each trial of this task, the participants were required to arrange ≤60 words (nouns or adjectives) in a two-dimensional space according to their semantic relationship on a computer screen by mouse drag-and-drop operations (Fig 2). This paradigm has allowed for the efficient collection of the perceptual semantic dissimilarity between words from participants [19].

The words used in this task included 60 nouns and 60 adjectives in Japanese (Table 3). These words were selected from the vocabulary of the fastText space (i.e., top 100,000 most frequently used words in the Japanese Wikipedia corpus). Nouns were selected in terms of the following six semantic categories: humans, non-human animals, non-animal natural things,

**Table 3. Words used in psychological experiments and for calculating word dissimilarity matrices.**

| Nouns | Adjectives |
|---|---|
| Employee, carpenter, teacher, student, driver, player, actor, writer, doctor, researcher, monkey, dog, cat, horse, cow, sheep, bird, frog, fish, bug, star, sky, mountain, sea, river, water, flower, tree, dirt, vegetable, house, school, factory, hospital, station, airport, bridge, hotel, stadium, park, car, motorbike, train, ship, boat, airplane, helicopter, rocket, bicycle, carriage, movie, book, music, phone, PC, TV, cloth, table, cup, text | Many, high, strong, large, absent, good, near, long, new, few, deep, broad, low, young, bad, bold, small, short, red, early, white, beautiful, black, old, weak, intense, far, detailed, childish, tender, bright, quick, correct, strict, blue, slow, sweet, thick, narrow, difficult, dim, rare, thin, heavy, light, pleasant, slender, wide, dark, hot, scary, shallow, sharp, circular, interesting, pale, sad, equal, remarkable, terrific |

Original words were in Japanese, and, for display purposes, here they were translated into English.

constructs, vehicles, and other artifacts. Ten words were selected for each category. For adjectives, such category-based selection was deemed difficult due to the small number of adjectives in the vocabulary (only 473 words). Nouns and adjectives were separately used in two distinct sessions of the task, which was performed over 2 days.

The words were arranged in a designated circular area on the computer screen ("arena"). The words were initially displayed outside the arena. The participants used mouse drag-and-drop operations in moving each word item into the arena and arranging them. The arrangement of words was according to the participants' own judgment of semantic similarity and dissimilarity between words. More similar or dissimilar word pairs should be closer or further apart in the arena, respectively. The participants were allowed to move any words within the arena as many times as they wished. Once all words were moved into the arena, the participants could click a button marked "Next" anytime to move on to the next trial.

On the initial trial of each session, the participants have arranged the entire set of 60 words (nouns or adjectives). On subsequent trials, they arranged subsets of those 60 words. After the end of each trial, the rough estimate of the word dissimilarity matrix and the evidence (0–1) for each word pairwise dissimilarity were computed as described previously [19]. Words in the subsets chosen for each trial were determined so as to increase evidence for pairwise dissimilarities of the words whose evidence estimated on the previous trial was the weakest. The session continued until the evidence of every word pair dissimilarity was above a threshold (0.75) or until the total duration of the session approached 1 h.

## Word dissimilarity matrix

The brain-derived word dissimilarity matrix for the word vector-based models was estimated using the following procedure: First, the $4K \times N$ voxelwise model weights for each participant were transformed into $K \times N$ by averaging the weights across the four sets of hemodynamic delay terms. Second, all $N$ or top $M$ voxels with the highest prediction accuracy were then selected from the model weights. All voxels were used in the main analysis, and selected voxels ($M$ = 2000, 5000, 10000, 30000, and 50000) were used in an additional analysis that tested the effect of voxel selection on brain–behavior correlation. Third, the $K$-dimensional fastText, GloVe, or word2vec vectors for the words used in the psychological experiments (Table 3) were multiplied by the $K \times N$ weight matrix, which yielded $N$- or $M$-dimensional word representations in the modeled brain space. Finally, a word dissimilarity matrix was calculated by the correlation distance (1 –Pearson's correlation coefficient) between these word representations, which was averaged across participants separately for nouns and adjectives.

The brain-derived word dissimilarity matrix for the voxelwise model based on binary labels (binary-labeling model) was also estimated. For the binary-labeling model, the $K \times N$ matrix

into which the $4K{\times}N$ model weights were transformed by averaging across hemodynamic delay terms corresponded to the brain representations of $K$ words. Accordingly, a word dissimilarity matrix of this model was calculated by the correlation distance between these $N$-dimensional word representations. Note that unlike the word vector-based models, the representations for the binary-labeling model contained only a limited part of the words used in the psychological experiments (Table 3). The number of words contained depended on vector dimensionality and movie sets (S18 Table).

The behavior-derived word dissimilarity matrix was evaluated from multiple word-subset arrangements on the word-arrangement task. The dissimilarity for a given word pair was estimated as a weighted average of the scale-adjusted dissimilarity estimates from individual arrangements as has been described previously [19]. These estimates produced a word dissimilarity matrix for the entire set of words. In this way, the word dissimilarity matrix was estimated separately for nouns and adjectives and was averaged across participants.

In addition, a word vector-derived word dissimilarity matrix was also examined in order to test whether the behavioral correlates of semantic dissimilarity structures changed through the transformation from original word vector representations to brain word representations. The word vector-derived matrix was computed by the correlation distance between fastText, GloVe, or word2vec vectors of the same word sets (Table 3) separately for nouns and adjectives.

The behavioral correlates of brain- or word vector-derived data were evaluated using Spearman's correlation between the brain- and behavior-derived word dissimilarity matrices (brain–behavior correlation) and between the word vector- and behavior-derived word dissimilarity matrices (word vector–behavior correlation). The upper triangular portion of each matrix was used for calculating correlation coefficients. For the binary-labeling model, correlation coefficients were calculated using only the words contained in the vocabulary of the model (S18 Table).

A permutation test was performed to evaluate the statistical significance of brain–behavior and word vector–behavior correlations. In each repetition of this test, the rows of a brain- or word vector-derived word dissimilarity matrix were randomly shuffled. Then, this shuffled brain- or word vector-derived matrix was used to calculate a Spearman's correlation with the original behavior-derived matrix. This procedure was repeated 10,000 times to obtain a null distribution of correlation coefficients and estimate the p value of an actual correlation coefficient. The difference of correlation coefficients between different matrix pairs was also tested in a similar manner. In this case, the difference of brain–or word vector–behavior correlation coefficients between different matrix pairs was calculated in each repetition while a brain- or word vector-derived matrix was shuffled separately for each of the different matrix pairs. This procedure was repeated 10,000 times to estimate a null distribution of the correlation coefficient differences.

## Supporting information

**S1 Fig. Model performance comparison across fastText vector dimensions. A**) Mean prediction accuracy of voxelwise models based on the fastText vector space with different vector dimensions. Error bars indicate standard error of the mean (SEM). **B**) Difference of mean prediction accuracies between different vector dimensions. The difference was evaluated separately for movie sets 1 (left) and 2 (right). The color of each cell represents the accuracy difference of the dimension on the x-axis minus the dimension on y-axis (red, positive values; blue, negative values). The mark in each cell indicates the statistical significance of the difference (Wilcoxon test, ${}^{***}p < 0.0001$, ${}^{**}p < 0.01$, ${}^{*}p < 0.05$, n.s., $p > 0.05$, FDR corrected). **C**)

Fractions of significant voxels for voxelwise models with different vector dimensions. The same conventions were used as in **A**. **D)** Difference in fractions of significant voxels between different vector dimensions. The same conventions were used as in **B**.
(TIF)

**S2 Fig. Model performance comparison across GloVe vector dimensions.** The same analysis as in S1 Fig but for GloVe vectors.
(TIF)

**S3 Fig. Model performance comparison across word2vec vector dimensions.** The same analysis as in S1 Fig but for word2vec vectors.
(TIF)

**S4 Fig. Comparison of prediction performance with voxelwise models based on discrete word features.** To compare discrete word features with word vectors in terms of the modeling of brain semantic representations, voxelwise models with the binary labeling of movie scenes (binary-labeling models) were constructed for individual participants. Prediction accuracy (top) and the fraction of significant voxels (bottom) for the binary-labeling models (pink bars) are shown separately for the vector dimensionality of 100, 300, 500,1000, and 2000 (from left to right), along with prediction performance for the word vector-based models (green bars, fastText; orange bars, GloVe; blue bars, word2vec). Error bars indicate SEM. Marks above bars indicate the statistical significance of the performance difference between the binary-labeling model and each of the word vector-based models (Wilcoxon test, $^{***}p < 0.0001$, $^{**}p < 0.01$, $^{*}p < 0.05$, FDR corrected).
(TIF)

**S5 Fig. Comparison of model prediction performance between trained and untrained word vectors.** To test whether the semantic relational structure of words, captured by word vectors, is effective in the modeling of cortical semantic representations, we compared trained (original) and untrained word vectors in terms of the model performance of brain-response prediction. We obtained untrained vectors by randomly assigning a 1000-dimensional random vector to each word in the same vocabulary as used for trained vectors. Hence, untrained vectors had signatures of individual words but not the semantic relational structure of words. Voxelwise models based on untrained vectors were constructed using the same procedure as used for voxelwise models based on trained vectors. **A)** Prediction accuracy for models based on trained vectors (x-axis) and ones based on untrained vectors (y-axis) separately shown for each word-vector type (top, fastText; middle, GloVe; bottom, word2vec) and each dataset (left, movie set 1; right, movie set 2). Each dot represents mean prediction accuracy over all voxels of each brain. The models based on trained vectors exhibited significantly higher prediction accuracy than those based on untrained vectors regardless of vector types and datasets (Wilcoxon test, $p < 0.00001$, FDR corrected). **B)** The fraction of significant voxels for models based on trained vectors (x-axis) and ones based on untrained vectors (y-axis). Each dot represents the fraction for each brain. The models based on trained vectors exhibited significantly higher fraction than those based on untrained vectors regardless of vector types and datasets ($p < 0.00001$, FDR corrected).
(TIF)

**S6 Fig. Cortical mapping of prediction accuracy for other dimensions of fastText vectors.** The mean prediction accuracy of fastText vector-based voxelwise models in each brain region was mapped onto the cortical surface (**A**, vector dimension = 100; **B**, 300; **C**, 500; **D**, 2000). The same conventions were used as in Fig 4. The five numbered cortical regions with the

highest mean prediction accuracy for each dimension and dataset are shown in S1 Table.
(TIF)

**S7 Fig. Cortical mapping of prediction accuracy for other dimensions of GloVe vectors.**
The same analysis as in S6 Fig but for GloVe vector-based models. The numbered cortical
regions are shown in S2 Table.
(TIF)

**S8 Fig. Cortical mapping of prediction accuracy for other dimensions of word2vec vectors.**
The same analysis as in S6 Fig but for word2vec vector-based models. The numbered cortical
regions are shown in S3 Table.
(TIF)

**S9 Fig. Cortical mapping of prediction accuracy at the voxel level.** Voxelwise prediction
accuracy of each word vector-based voxelwise model (vector dimension = 1000) for a repre-
sentative participant was mapped onto the cortical flat map of the participant (from top to bot-
tom: fastText, GloVe, word2vec, and binary labeling models; left: movie set 1, right: movie set
2). Brighter colors on the cortical maps indicate voxels with higher prediction accuracy. Only
voxels with prediction accuracy above 0.10 are shown. The values of voxelwise prediction
accuracy range up to 0.709. White lines on the cortical maps denote representative sulci: CoS,
collateral sulcus; STS, superior temporal sulcus; TOS, transverse occipital sulcus; IPS, intrapar-
ietal sulcus; SyF, sylvian fissure; CeS, central sulcus; SFS, superior frontal sulcus; IFS, inferior
frontal sulcus.
(TIF)

**S10 Fig. Upper range of voxelwise prediction accuracy.** The distribution of the maximum
values of voxelwise prediction accuracy for individual participants was shown separately for
each model (**A** and **B**: fastText; **C** and **D**: GloVe; **E** and **F**: word2vec; **G** and **H**: binary-labeling)
and each dataset (**A**, **C**, **E**, and **G**: movie set 1; **B**, **D**, **F**, and **H**: movie set 2). The vertical dashed
line in each panel indicates the mean value of the maximum prediction accuracy averaged
across participants.
(TIF)

**S11 Fig. Cortical mapping of the fraction of significant voxels for other dimensions of fas-
tText vectors.** The mean fraction of significant voxels in each brain region for fastText vector-
based models was mapped onto the cortical surface (**A**, vector dimension = 100; **B**, 300; **C**,
500; **D**, 2000). The same conventions were used as in Fig 5. The five numbered cortical regions
with the highest mean fraction of significant voxels for each dimension and dataset are shown
in S7 Table.
(TIF)

**S12 Fig. Cortical mapping of the fraction of significant voxels for other dimensions of
GloVe vectors.** The same analysis as in S11 Fig but for GloVe vector-based models. The num-
bered cortical regions are shown in S8 Table.
(TIF)

**S13 Fig. Cortical mapping of the fraction of significant voxels for other dimensions of
word2vec vectors.** The same analysis as in S11 Fig but for word2vec vector-based models. The
numbered cortical regions are shown in S9 Table.
(TIF)

**S14 Fig. Comparison of the fraction of significant voxels between original and control
word vector-based models.** To test the significance of the fraction of significant voxels for

original models, we compared the original fraction with the fraction of significant voxels for control models. The control models used word vectors randomly shuffled across vector dimensions for each vector and thereby produced the chance level of the fraction of significant voxels. The fraction of significant voxels in each cortical region for these original (y-axis) and control (x-axis) models is shown separately for each word-vector type (**A** and **B**: fastText; **C** and **D**: GloVe; **E** and **F**: word2vec) and each dataset (**A**, **C**, and **E**: movie set 1; **B**, **D**, and **F**: movie set 2). Each dot represents the fraction in each cortical region. Regardless of the word-vector types and datasets, the fraction for the original models was significantly higher than that for the control models in all the cortical regions (Wilcoxon test, $p < 0.00001$, FDR corrected). This result indicates that the fraction of significant voxels for the original models is sufficiently above chance level.
(TIF)

**S15 Fig. Cortical mapping of prediction accuracy for binary-labeling models.** The same analysis as in S6 Fig but for binary-labeling models with the vector dimensionality of 1000 and 2000. The numbered cortical regions are shown in S13 Table.
(TIF)

**S16 Fig. Cortical mapping of the fraction of significant voxels for binary-labeling models.** The same analysis as in S11 Fig but for binary-labeling models with the vector dimensionality of 1000 and 2000. The numbered cortical regions are shown in S14 Table.
(TIF)

**S17 Fig. Comparison of model prediction performance in each cortical region between trained and untrained word vectors.** Prediction performance of voxelwise models (**A**, prediction accuracy; **B**, the fraction of significant voxels) in each cortical region was compared between trained (x-axis) and untrained word vectors (y-axis). The performance is shown separately for each word-vector type (top, fastText; middle, GloVe; bottom, word2vec) and each dataset (left in each of **A** and **B**, movie set 1; right, movie set 2). Each dot represents the mean performance averaged over participants for each region. The difference of model prediction performance between trained and untrained vectors was tested within each region while the performance for each participant was used as a data sample. Filled and open dots indicate that the region-wise difference was significant or not, respectively (Wilcoxon test, $p < 0.05$, FDR corrected). Then, the difference of model prediction performance between trained and untrained vectors was tested across regions while the mean performance in each region was used as a data sample. The difference across regions were significant regardless of prediction performance measures, word-vector types, and datasets (Wilcoxon test, $p < 0.00001$, FDR corrected).
(TIF)

**S18 Fig. Correlations between dissimilarity matrices for all dimensions of fastText vectors.** **A**) Brain–behavior and word vector–behavior correlations of noun dissimilarity matrices for different dimensions of fastText vectors. **B**) Difference between brain–behavior correlation coefficients of different vector dimensions for nouns. The color of each cell represents the coefficient difference of the dimension on the x-axis minus the dimension on the y-axis (red, positive values; blue, negative values). The mark in each cell indicates the statistical significance of the difference (permutation test, ***$p < 0.0001$, **$p < 0.01$, *$p < 0.05$, n.s., $p > 0.05$, FDR corrected). (**C** and **D**) The same analyses were performed as in **A** and **B** using adjective dissimilarity matrices.
(TIF)

**S19 Fig. Correlations between dissimilarity matrices for all dimensions of GloVe vectors.** The same analysis as in S18 but for GloVe vectors.
(TIF)

**S20 Fig. Correlations between dissimilarity matrices for all dimensions of word2vec vectors.** The same analysis as in S18 but for word2vec vectors.
(TIF)

**S21 Fig. Correlations between brain- and behavior-derived word dissimilarity matrices from the same participant population.** We constructed word dissimilarity matrices using behavioral and voxelwise-model data obtained from the same population of 6 participants for movie set 1. **A**, **B**) Behavior- and brain-derived dissimilarity matrices for nouns (**A**) and adjectives (**B**). The brain-derived matrices were obtained using 1000-dimensional fastText vectors. The same conventions were used as in Fig 6. **C**) Brain–behavior correlations for nouns and adjectives. All the correlation coefficients were significantly higher than chance level (permutation test, $p < 0.0001$, FDR corrected).
(TIF)

**S22 Fig. Brain–behavior correlations of word dissimilarity at the individual level.** We calculated the Spearman's correlation between brain- and behavior-derived word dissimilarity matrices for each of the 6 participants who had both brain and behavioral data. The correlation coefficients are shown separately for nouns (**A**) and adjectives (**B**) and for each type of word vectors (dimensionality = 1000). Marks above bars indicate the statistical significance of the correlation coefficients (permutation test, $^{***}p < 0.0001$, $^{**}p < 0.01$, $^{*}p < 0.05$, FDR corrected).
(TIF)

**S23 Fig. Difference of brain–behavior correlation coefficients between each pair of different voxel selections for fastText vector-based models.** The color of each cell represents the difference between each pair of the number of selected voxels; the difference is the brain–behavior correlation coefficient for the dimension on the x-axis minus the coefficient for the dimension on the y-axis (red, positive values; blue, negative values). The mark in each cell indicates the statistical significance of the difference (permutation test, $^{***}p < 0.0001$, $^{**}p < 0.01$, $^{*}p < 0.05$, FDR corrected).
(TIF)

**S24 Fig. Difference of brain–behavior correlation coefficients between each pair of different voxel selections for GloVe vector-based models.** The same analysis as in S23 Fig but for GloVe vectors.
(TIF)

**S25 Fig. Difference of brain–behavior correlation coefficients between each pair of different voxel selections for word2vec vector-based models.** The same analysis as in S23 Fig but for word2vec vectors.
(TIF)

**S26 Fig. Brain–behavior correlation for each noun category.** The nouns used for constructing word dissimilarity matrices consisted of the six categories (humans, non-human animals, non-animal natural things, constructs, vehicles, and other artifacts; 10 nouns in each category). We calculated the correlation between behavior- and brain-derived word dissimilarity matrices obtained from each noun category. The Pearson's correlation coefficients for each category were shown separately for each model (green, fastText; orange, GloVe; blue, word2vec) and

for each movie set (**A**, movie set 1; **B**, movie set 2). Marks above bars indicate the statistical significance of the correlation coefficients (permutation test, ***p < 0.0001, **p < 0.01, *p < 0.05, FDR corrected).
(TIF)

**S27 Fig. Comparison of brain–behavior correlation with binary-labeling models.** The correlation of brain- and behavior-derived word dissimilarity matrices was computed for binary-labeling models. The correlation coefficients for these models (pink bars) are separately shown for nouns (top) and adjectives (bottom) and for the vector dimensionality of 300, 500, 1000, and 2000 (from left to right), along with the coefficients of brain–behavior correlation for the word vector-based models (green bars, fastText; orange bars, GloVe; blue bars, word2vec). In this case, because the vocabulary of the binary-labeling models changed depending on vector dimensionality and datasets (S16 Table), the number of nouns and adjectives used for calculating the brain-behavior correlation of both binary-labeling and word vector-based models were different across vector dimensions and datasets. Consequently, the correlation coefficients for the word vector-based models differed from those shown in other figures (Figs 6–7 and S18–S20). Marks below bars indicate the statistical significance of correlation coefficients above chance level whereas marks above bars indicate the statistical significance of the correlation difference between the binary-labeling model and each of the word vector-based models (permutation test, ***p < 0.0001, **p < 0.01, *p < 0.05, FDR corrected).
(TIF)

**S1 Table. Cortical regions with the highest mean prediction accuracy for each of the other dimensional fastText vectors.**
(TIFF)

**S2 Table. Cortical regions with the highest mean prediction accuracy for each of the other dimensional GloVe vectors.**
(TIFF)

**S3 Table. Cortical regions with the highest mean prediction accuracy for each of the other dimensional word2vec vectors.**
(TIFF)

**S4 Table. Consistency of inter-regional patterns of prediction accuracy for fastText vectors across vector dimensionality and across movie sets.** We calculated the mean prediction accuracy of fastText vector-based models within each cortical region (Figs 4 and S6) and compared the inter-regional patterns of prediction accuracy between arbitrary pairs of vector dimensionality and movie sets. The value in each cell in the upper and lower part of the table denotes the Pearson's or Spearman's correlation coefficient, respectively, of the inter-regional patterns between each pair.
(TIFF)

**S5 Table. Consistency of inter-regional patterns of prediction accuracy for GloVe vectors.** The same analysis as in S4 Table but for GloVe vectors.
(TIFF)

**S6 Table. Consistency of inter-regional patterns of prediction accuracy for word2vec vectors.** The same analysis as in S4 Table but for word2vec vectors.
(TIFF)

**S7 Table. Cortical regions with the highest mean fraction of significant voxels for each of the other dimensional fastText vectors.**
(TIFF)

**S8 Table. Cortical regions with the highest mean fraction of significant voxels for each of the other dimensional GloVe vectors.**
(TIFF)

**S9 Table. Cortical regions with the highest mean fraction of significant voxels for each of the other dimensional word2vec vectors.**
(TIFF)

**S10 Table. Consistency of inter-regional patterns of significant-voxel fraction for fastText vectors across vector dimensionality and across movie sets.** As in the case of the mean prediction accuracy (S4 Table), the inter-regional patterns of significant-voxel fraction of fastText vector-based models (Figs 5 and S11) were compared between arbitrary pairs of vector dimensionality and movie sets. The value in each cell in the upper and lower part of the table denotes the Pearson's or Spearman's correlation coefficient, respectively, of the inter-regional patterns between each pair.
(TIFF)

**S11 Table. Consistency of inter-regional patterns of significant-voxel fraction for GloVe vectors.** The same analysis as in S10 Table but for GloVe vectors.
(TIFF)

**S12 Table. Consistency of inter-regional patterns of significant-voxel fraction for word2-vec vectors.** The same analysis as in S10 Table but for word2vec vectors.
(TIFF)

**S13 Table. Cortical regions with the highest mean prediction accuracy for each of 1000- and 2000-dimensional binary-labeling models.**
(TIFF)

**S14 Table. Cortical regions with the highest mean fraction of significant voxels for each of 1000- and 2000-dimensional binary-labeling models.**
(TIFF)

**S15 Table. Cortical regions with the highest brain–behavior correlations of noun dissimilarity for each word-vector type.**
(TIFF)

**S16 Table. Cortical regions with the highest brain–behavior correlations of adjective dissimilarity for each word-vector type.**
(TIFF)

**S17 Table. Number of stimulus movie clips in individual product/service categories.**
(TIFF)

**S18 Table. Number of words contained in the vocabulary of the binary-labeling model.**
(TIFF)

## Acknowledgments

We thank Ms. Hitomi Koyama, Mr. Yusuke Nakano, Mr. Koji Takashima, Mr. Takeshi Matsuda, Mr. Susumu Minamiyama, Ms. Mami Yamashita, Mr. Ryo Yano, Ms. Risa Matsumoto, Mr. Masato Okino, and Mr. Akira Nagaoka for their analytical and experimental support.

## Author Contributions

**Conceptualization:** Satoshi Nishida.

**Data curation:** Satoshi Nishida, Antoine Blanc, Naoya Maeda, Masataka Kado.

**Formal analysis:** Satoshi Nishida, Antoine Blanc.

**Funding acquisition:** Satoshi Nishida, Shinji Nishimoto.

**Investigation:** Satoshi Nishida.

**Methodology:** Satoshi Nishida, Antoine Blanc.

**Project administration:** Satoshi Nishida.

**Resources:** Satoshi Nishida, Naoya Maeda, Masataka Kado.

**Software:** Satoshi Nishida, Antoine Blanc.

**Supervision:** Satoshi Nishida, Shinji Nishimoto.

**Validation:** Satoshi Nishida.

**Visualization:** Satoshi Nishida.

**Writing – original draft:** Satoshi Nishida.

**Writing – review & editing:** Satoshi Nishida, Antoine Blanc, Naoya Maeda, Masataka Kado, Shinji Nishimoto.

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
