## [Decision Letter · Decision Letter 0]

1 Dec 2020

Dear Dr. Nishida,

Thank you very much for submitting your manuscript "Behavioral correlates of cortical semantic representations modeled by word vectors" for consideration at PLOS Computational Biology.

As with all papers reviewed by the journal, your manuscript was reviewed by members of the editorial board and by several independent reviewers. In light of the reviews (below this email), we would like to invite the resubmission of a significantly-revised version that takes into account the reviewers' comments.

Dear Dr. Nishida and co-authors,

I am very glad to handle this submission. Now three reviews have returned for the manuscript. Overall, reviewers found the paper interesting. However, they also raised important issues. I agree with the reviewers that these issues should be addressed before the paper can be considered for acceptance.

One concern is novelty. Two reviewers worry that the paper has not provided significant new insights and have pointed out relevant literature that should be compared to explain the novelty more clearly.

Another is the validity of methodology and sufficiency of analysis. Reviewer 1 and 2 both asked for additional analysis that can help readers better evaluate the conclusion. Clarification of some of the analysis is needed before they can assess the validity of the claims made. I agree with all these and think they should be carefully addressed. When I read the manuscript, I had the same question as the second point in concern (2) of reviewer 1: what the neural similarity would look like if simple binary labels of the existence of object corresponding to the nouns are used to build design matrix and estimate neural patterns. Please also address other issues raised.

I noticed that fraction of significant voxels, accuracy, dissimilarity matrix have been provided in the data repository which allows plotting the figures in the manuscript. However, such quantities still do not allow readers to reproduce the findings (including these derived quantities themselves) without the data underlying them. For example, several questions of Reviewers #1 and #2 cannot be answered based on these derived quantities. PLOS journals require authors to make all data necessary to replicate their study’s findings publicly available without restriction at the time of publication. Please follow the Data Availability policy of PLOS (https://journals.plos.org/ploscompbiol/s/data-availability) to provide data that allows reproduction of the findings in the manuscript.

We cannot make any decision about publication until we have seen the revised manuscript and your response to the reviewers' comments. Your revised manuscript is also likely to be sent to reviewers for further evaluation.

Sincerely,

Ming Bo Cai

Associate Editor

PLOS Computational Biology

Lyle Graham

Deputy Editor

PLOS Computational Biology

Reviewer's Responses to Questions

**Comments to the Authors:**

Reviewer #1: Summary

The authors investigate the extent to which artificial models of language (NLP) can predict human neural responses and align with human behavioral judgments. They find significant agreement between brain, model, and behavior and conclude that current word embedding models are able to capture a meaningful amount of variance of both neural representation and semantic perception in humans.

Overall Impressions

I think this question is very interesting and I think that progress in this domain would be very welcome and useful for everyone who is interested in semantics, neural correlates of perception, and/or prediction of neural representations (myself included). I also like the approach (which is consistent with prior work), but I’m not fully convinced that the present results provide significant new insights into the problem in their current form. My main concerns about the current manuscript relate to the choice of embedding model, the lower effect sizes compared to similar prior work, and not fully accounting for known neural biases in how the effects are analyzed. However, I believe that these concerns could be fully addressed by additional analyses and discussion (as detailed below).

Major Comments

(1) The embedding model selected has weaker performance than openly available alternatives: Pereira et al. 2018 shows that multiple skip-gram models (e.g., word-2-vec, which is purportedly not as ‘good’ as fastText) can predict concept similarity up to r~=0.70. There’s also some recent work from Iordan et al. 2019 that pushes this to r~=0.90 for certain categories. In the current work, fastText is only at r~=0.30 with behavior and the authors argue that this means the brain mapping helps (r=~0.60), but it may be that other word embedding models can already do much better than any brain mapping at predicting behavior for this particular set of adjectives and/or nouns, but that the authors did not pick a good enough embedding model. To provide evidence that this is not the case, the authors could (1) test other embedding models listed above and show that they have similar behavioral performance for their set of concepts; and/or (2) show that at least one such other model (especially if it outperforms fastText) can also give improved predictions when used to predict neural representations similar to the current analysis.

(2) Neural pattern prediction accuracy is much lower and spatially different from prior work: The pattern of model performance (Fig. 4, top) is very different from that of prior work and performance is much lower (i.e., r~=0.10-0.30 compared to e.g. Fig. 1 of Huth et al. 2016, r=0.60 in high-level regions). Spatially, in the present work, prediction is highest in LO, which is below chance in the Huth language model; also, current model has little to no significant frontal cortex predictions. One could argue that this is because Huth et al. 2016 tries to use a language model to predict auditory stories, not movies. But then, performance of the current model is even weaker when compared to Huth et al. 2012 (r values up to 0.80) which uses simple binary labelling of the objects occurring in videos to predict neural response. Hence, I am not convinced that this language model is the best approach to model visual content, e.g., complex movies. To provide evidence that this is a good/comparable/better alternative than existing methods, the authors could use their current data to automatically generate a binary matrix of objects (similar to Huth et al. 2012) for every timepoint and use this to perform neural prediction, instead of using the word vector representations for the objects/concepts. If this matches/improves performance, then this could show evidence that the word vector approach either doesn’t add anything or may even hinder recovery of semantic information by introducing external biases.

(3) Correlations don’t control for the known animate/inanimate distinction in VTC: One of the most salient distinctions in patterns of activity in occipito-temporal cortex is between animate and inanimate stimuli/categories. From the list on line 522, I see a clear human-animal-manmade clustering of nouns which is also consistent with the behavioral matrix in Fig. 4. If the model simply picks up on the highest variance dimension of representation across cortical patterns (e.g., animacy), then this could drive most of the correlation effects observed. However, we already know this animacy effect is present in the brain (Connolly et al. 2012, Konkle & Caramazza 2013, Kriegeskorte et al. 2008, etc.), so, in this case, this model would tell us nothing new about semantic knowledge representation in the brain. To address this issue, the authors could try to characterize and control for the effects of animacy (and the human-animal distinction within that) when measuring how well the model predicts behavior and/or the brain (i.e., regress this distinction out or focus their analysis on similarity matrices within these superordinate categories). If the effects survive, then this would indeed tell us that the models capture something interesting and novel about the semantic structure of neural representations and their connections with behavioral measures / perception, beyond animacy.

Minor Comments

(4) line 145: point to the corresponding supplementary figure (S5?). Authors could also consider running a Friedman test to assess non-monotonicity statistically.

(5) line 151: how do you define ‘appropriate’? It would be helpful if you mentioned briefly in the text that you used Freesurfer anatomical segmentation.

(6) Fig. 4: it would be helpful to have labels for these brain regions in the caption / text / supplement. Is the yellow one in a/b LO?

(7) Fig. 4: A voxel-wise map similar to Huth et al. 2012/2016 would be helpful here to compare how this method can model semantic information in the brain (i.e., same figure, but with voxel-level granularity, not just Freesurfer ROI).

(8) line 273: the effect sizes are really low (1-2% neural variance explained) and the authors are over-inflating the value of the procedure here.

(9) It would be useful to discuss why you chose fastText over other models. This could also be addressed in conjuction with Major Comment #1 above.

(10) Multiband factor 6 is quite high and the TR=1s is quite fast -- this may cause very low SNR (which maybe would explain low r values compared to similar work?). Is there a reason you did not use a more conventional fMRI sequence (e.g., multiband 2/4, 1.5-2s TR)?

(11) line 228: The authors should be a little more careful with the strength of the conclusions they draw from their results.

(12) line 310: I don’t agree with the argument that the language regions are not involved in processing this type of movie info – I suspect that an activation GLM analysis on the movie data would show weak, but consistent activation in language regions that the current model cannot capture.

(13) line 330: I don't necessarily agree that word embeddings are more explainable than deep network representations; they are just as opaque. I tried to find work supporting the authors’ claim but couldn’t. Please provide some references here and expand on this explanation or please reword.

Reviewer #2: I have now had a chance for an in-depth review of “Behavioral correlates of cortical semantic representations modeled by word vectors” by Nishida et al. The manuscript is well written and the overall problem domain is derived clearly. In this project, the authors use encoding models based on a FastText embedding space to predict fMRI brain data and subsequently use the predicted beta values to compute a representational dissimilarity matrix (RDM) for a novel set of words. The resulting RDM is compared to a behavioural RDM obtained via inverse MDS on the same words. Using this elaborate pipeline, the authors come to the conclusion that the predicted brain data and behavioural judgments are correlated, indicating the feasibility of using semantic embedding spaces for brain and behavioural modelling.

The overall approach is interesting, especially since the brain data was obtained via a visual/sensory modality, whereas the encoding model is derived from more abstract textual scene descriptions. To my understanding this cross-domain encoding approach acts as a “filtering” function that highlights brain regions related to semantics rather than low-level visual brain regions. This strength of the paper could have been highlighted a bit more. As stated above, the paper is well written, and computationally elegant. However, I have a few comments, as detailed below, which I hope will help the authors improve their work.

Major 1: Take-home message and novelty

While I see the elegance and computational complexity of the author’s endeavour, I am a bit lost as to what the actual take home message of the work may be and where and how the authors advance beyond what is known about the brain from prior work. I therefore invite the authors to elaborate on this.

At multiple occasions, the authors underline the novelty of the work, stating that “[…] there has been no study explicitly examining whether the modeled semantic representations actually capture our perception of semantic information.” and “However, whether the semantic representations modeled by word vectors accurately reflect the semantic perception of humans is yet to be determined.". However, there is a large body of research investigating exactly these questions (see [1-6] for examples of previous work most of which remain uncited).

Major 2: Technical validity

Going through the manuscript, I found myself continuously wondering about the exact statistical procedures run for the different tests described. This made it quite difficult to judge the actual merit of the claims made. Below are examples of cases that need technical clarification/adjustments:

a. The authors write “the fraction of significant voxels that reach their prediction accuracy to a threshold value”. It remains unclear, however, how significance was established, what the threshold value was, what statistical test was used, which data was tested, etc. As a second example, the authors write “Only regions with mean prediction accuracy above 0.11 (corresponding to p = 0.0001 ~ 0.05/150 regions) are shown”. Can the authors clarify what exactly they did here to derive the statistics?

b. The minimum fraction of significant voxels within any individual region is reported as 0.237. This seems very high, given that the average model prediction accuracy across all voxels is reported as “0.09”. To better understand this discrepancy, I would like the authors to comment on (a) how exactly the test set was derived, (b) how exactly the authors tested for significance (see above comment), and (c) how many training/test datapoints were used. Secondly, testing against chance, after having fitted hundreds of free parameters, seems like a somewhat low bar (although I acknowledge the fact that this is prediction on test data). As a more conservative control, I suggest the authors use an untrained embedding network as a control model as this would give the claims for actual semantic embedding throughout the cortex more explanatory power/merit.

c. “There was a strong correlation of mean prediction accuracy over 150 cortical regions between movie sets 1 and 2 (Pearson’s r = 0.974);“ Please repeat this analysis with a robust Spearman correlation, as deviations from normality (such as outliers) can easily increase the Pearson correlation coefficient but this is not meaningful. Please note that nearby regions are autocorrelated, which may violate the independence assumption of the classic Pearson statistic.

d. To my understanding, the significance of correlations between RDMs (Figure 5) is computed via bootstrapping individual RDM cells. I do not think that this is technically valid, as the RDM cells have dependencies that are violated by this approach (the N*N-1 cells are all derived from only N experimental conditions). The more standard approach, which I advise the authors to take, would be to perform a permutation test on the RDM conditions (i.e. permuting the rows/columns of one of the two RDMs to derive a null distribution against which the empirical correlation value can be tested). Moreover, Spearman correlations are more commonly used for RDM comparisons across domains.

Minor:

1. Which exact atlas was used? This is not mentioned anywhere. Relatedly, were the 150 regions tested all regions of the atlas, or were ROIs excluded?

2. Please consider citing [7], which is a novel paper in the domain.

3. Could the authors attempt a spatial localization of brain-behavioural semantic processing by computing RDMs for each ROI and correlate it with the behavioural data?

I sincerely hope that the above comments will be perceived as constructive and that they may help assist the authors in making their contribution stronger. It is an exciting project and overall computational approach.

Signed

Tim Kietzmann

[1] https://arxiv.org/pdf/1910.06954.pdf

[2] Mikolov, T., Yih, S. W. & Zweig, G. Linguistic Regularities in Continuous Space Word Representations. In Proceedings of the 2013 Conference of the North American Chapter of the Association for Computational Linguistics: Human Language Technologies, 746-751 (2013).

[3] https://arxiv.org/abs/1408.3456

[4] Pereira, F., Gershman, S., Ritter, S. & Botvinick, M. A comparative evaluation of off- the-shelf distributed semantic representations for modelling behavioural data. Cogn. Neuropsychol. 33, 175–190 (2016).

[5] https://pubmed.ncbi.nlm.nih.gov/29777825/

[6] https://arxiv.org/pdf/1805.07644.pdf

[7] https://www.nature.com/articles/s41562-020-00951-3

Reviewer #3: Although a number of studies demonstrated correspondences between brain activity and word embeddings, the behavioral validation of these studies has been lacking. The authors address this issue by using representational similarity analysis on behavioral data and fMRI data in order the compare the two, revealing a significant correlation. It is a clearly written manuscript with well-designed experiments and analyses. I really liked that it has a clear message and delivers it well without going into tangential research questions which would distract from the main results. I would be happy to recommend accepting the manuscript for publication after the few minor issues that I list below are addressed.

- It is not immediately clear how many participants were involved in the study and which tasks each participant performed. Although this information is given later in the methods section, it appears rather late in the manuscript. A table or a few sentences to clearly state this earlier in the manuscript could make it easier for the reader. On another note, just out of curiosity, I see that the number of participants was quite large in comparison to what we are used to seeing in similar studies. Was there a specific motivation for collecting such a large fMRI dataset?

- Not a big deal, but in Figure 4, the colors that are overlaid on to the brain surfaces are transparent, which makes it a bit confusing to read the results with the brain surface also having brightness differences. Making the results opaque would be preferable, in my opinion. At the same time, I see the importance of conveying the anatomical information to the reader, so I leave it to the authors to decide what to do with this comment.

- How was the variability of the RDMs between participants? Have the analyses been performed also at the individual level (e.g. comparing each participant’s behavioral data with their own brain data)? If not, why not?

**Have all data underlying the figures and results presented in the manuscript been provided?**

Reviewer #1: Yes

Reviewer #2: Yes

Reviewer #3: Yes

PLOS authors have the option to publish the peer review history of their article (what does this mean?). If published, this will include your full peer review and any attached files.

Reviewer #1: No

Reviewer #2: **Yes: **Tim C Kietzmann

Reviewer #3: No
---

## [Decision Letter · Decision Letter 1]

16 May 2021

Dear Dr. Nishida,

Thank you very much for submitting your manuscript "Behavioral correlates of cortical semantic representations modeled by word vectors" for consideration at PLOS Computational Biology. As with all papers reviewed by the journal, your manuscript was reviewed by members of the editorial board and by several independent reviewers. The reviewers appreciated the attention to an important topic. Based on the reviews, we are likely to accept this manuscript for publication, providing that you modify the manuscript according to the review recommendations.

We are very glad to have received your revision. Reviewers are generally satisfied with the updated version. There are still a few comments from Reviewer #2 that need addressing. It would be great if you can improve the manuscript based on these sugestions.

Sincerely,

Ming Bo Cai

Associate Editor

PLOS Computational Biology

Lyle Graham

Deputy Editor

PLOS Computational Biology

[LINK]

We are very glad to have received your revision. Reviewers are generally satisfied with the updated version. There are still a few comments from Reviewer #2 that need addressing. It would be great if you can improve the manuscript based on these sugestions.

Reviewer's Responses to Questions

**Comments to the Authors:**

Reviewer #1: I now had a chance to carefully read the new version and the responses, and I think the authors did a really nice job on the revisions. I would be happy to see this paper published.

Reviewer #2: I thank the authors for considering my earlier comments. I am happy with the current version of the manuscript. Below are some minor points that I think would further improve the paper (especially point 1).

Minor:

1. In response to one of my earlier comments, the authors derived a random control model (“we emulated an untrained embedding network […] using word vectors for which the vector representations were shuffled across vector dimensions for each word.). This approach seems overly lenient, as no sensible betas can be derived for the encoding model if the dimensions are shuffled randomly for each datapoint. I therefore do not see the immediate use in this control. What I had originally thought of, but perhaps not communicated precisely enough, was that the authors take an untrained DNN (rather than a semantics trained DNN) and run it through their pipeline. This ensures that the encoding model can make “sensible” predictions, while the model activation patterns themselves are not driven based on semantics, as the model is untrained.

2. Typo: Author summery —> Author summary

3. Figure 7 - would it make sense for the authors to add the binary category control model as an additional bar?

4. I would advise not calling binary noun/category descriptions “primitive” (l 678).

Reviewer #3: The authors have addressed all of my points in the revision. I have no more concerns left and would be happy to see the manuscript published in its current form.

**Have the authors made all data and (if applicable) computational code underlying the findings in their manuscript fully available?**

Reviewer #1: Yes

Reviewer #2: Yes

Reviewer #3: Yes

PLOS authors have the option to publish the peer review history of their article (what does this mean?). If published, this will include your full peer review and any attached files.

Reviewer #1: No

Reviewer #2: **Yes: **Tim C Kietzmann

Reviewer #3: No

Figure Files:

Data Requirements:

Reproducibility:

References:

---

## [Decision Letter · Decision Letter 2]

1 Jun 2021

Dear Dr. Nishida,

We are pleased to inform you that your manuscript 'Behavioral correlates of cortical semantic representations modeled by word vectors' has been provisionally accepted for publication in PLOS Computational Biology.

Best regards,

Ming Bo Cai

Associate Editor

PLOS Computational Biology

Lyle Graham

Deputy Editor

PLOS Computational Biology

Reviewer's Responses to Questions

**Comments to the Authors:**

Reviewer #2: The authors have addressed my remaining concerns. I suggest acceptance.

**Have the authors made all data and (if applicable) computational code underlying the findings in their manuscript fully available?**

Reviewer #2: None

PLOS authors have the option to publish the peer review history of their article (what does this mean?). If published, this will include your full peer review and any attached files.

Reviewer #2: No

---

## [Editor Report · Acceptance letter]

18 Jun 2021

PCOMPBIOL-D-20-01475R2 

Behavioral correlates of cortical semantic representations modeled by word vectors

Dear Dr Nishida,

I am pleased to inform you that your manuscript has been formally accepted for publication in PLOS Computational Biology. Your manuscript is now with our production department and you will be notified of the publication date in due course.

With kind regards,

Katalin Szabo
